# Alternate materials for the capture and quantification of gaseous oxidized mercury in the atmosphere

Livia Lown[1], Sarrah M. Dunham-Cheatham[2], Seth N. Lyman[3, 4], Mae S. Gustin[1]

[1]Department of Natural Resources and Environmental Science, University of Nevada, Reno, Reno, NV, 89557, USA
[2]College of Agriculture, Biotechnology & Natural Resources, University of Nevada, Reno, Reno, NV, 89557, USA
[3]Bingham Research Center, Utah State University, Vernal, UT, 84078, USA
[4]Department of Chemistry and Biochemistry, Utah State University, Logan, UT 84322, USA

Correspondence to: Mae S. Gustin (mgustin@unr.edu)

**Abstract.** Methodologies for identifying atmospheric oxidized mercury ($Hg^{II}$) compounds, including particulate-bound $Hg^{II}$ ($Hg^{II}_{(p)}$) and gaseous oxidized mercury ($Hg^{II}_{(g)}$), by mass spectrometry (MS) are currently under development. This method requires preconcentration of $Hg^{II}$ for analysis due to high instrument detection limits relative to ambient $Hg^{II}$ concentrations. The objective of this work was to identify and test materials for quantitative capture of $Hg^{II}$ from the gas phase, and to suggest potential surfaces onto which $Hg^{II}$ can be collected, thermally desorbed, and characterized using MS methods. From the literature, several compounds were identified as potential sorbent materials and tested in the laboratory for uptake of gaseous elemental mercury ($Hg^0$) and $Hg^{II}_{(g)}$ (permeated from a $HgBr_2$ salt source). Chitosan, $\alpha$-$Al_2O_3$, and $\gamma$-$Al_2O_3$ demonstrated $Hg^{II}_{(g)}$ capture in ambient air laboratory tests, without sorbing $Hg^0$ under the same conditions. When compared to cation exchange membranes (CEM), chitosan captured a comparable quantity of $Hg^{II}_{(g)}$, while $\leq 90\%$ of loaded $Hg^{II}_{(g)}$ was recovered from $\alpha$-$Al_2O_3$, and $\gamma$-$Al_2O_3$. When deployed in the field, the capture efficiency of chitosan decreased compared to CEM, indicating environmental conditions impacted the sorption efficiency of this material. The poor recovery of $Hg^{II}$ from the tested materials compared to CEM in the field indicate that further identification and exploration of alternative sorbent materials is required to advance atmospheric mercury chemistry analysis by MS methods.

## 1 Introduction

Mercury (Hg) is a toxic global contaminant that is introduced to aquatic and terrestrial ecosystems primarily from the atmosphere (Driscoll et al., 2013). Oxidized forms ($Hg^{II}$), including gaseous oxidized mercury ($Hg^{II}_{(g)}$) and particulate-bound mercury ($Hg^{II}_{(p)}$), are deposited from the atmosphere to ecosystems (Ariya et al., 2015), and may become available for transformation to and accumulation as methylmercury in food webs (Lyman et al., 2020a). A complete understanding of Hg behavior in the atmosphere is necessary to describe the fate of anthropogenic Hg pollution, assess health risks to humans and wildlife, and evaluate the effectiveness of the Minamata Convention. The mechanisms that govern the oxidation and reduction of Hg in the atmosphere are not well understood (Shah et al., 2021), and model results are uncertain, because they do not consider other forms of $Hg^{II}$ present (Gustin et al., 2023).

For example, Shah et al. (2021) assumed all $Hg^{II}$ compounds devolatilized from aerosols are $HgCl_2$. This is an assumption that has not been validated.

Currently, $Br^-$ and $Cl^-$ radicals are considered to participate in elemental Hg ($Hg^0$) oxidation. This is based on both theoretical work (Holms et al., 2010; Horowitz et al., 2017; Song et al., 2024) and experimental observations of atmospheric Hg depletion events in the Arctic (Steffen et al., 2008), as well as observations of $Hg^{II}$ formation at the marine boundary layer (Laurier et al., 2003). The identity of $Hg^{II}_{(g)}$ compounds in the atmosphere is currently unknown, but mass spectrometry (MS) methods capable of observing atmospheric $Hg^{II}_{(g)}$ speciation are in development (cf., Jones et al., 2016; Khalizov et al., 2020; Mao and Khalizov, 2021). These methods have been developed using $HgBr_2$ and $HgCl_2$ as model atmospheric $Hg^{II}_{(g)}$ compounds, given the role of halogen radicals in atmospheric $Hg^0$ oxidation. However, due to differences in $Hg^{II}_{(g)}$ behavior, the use of a broad range of representative compounds is desirable in both MS method development and validation of preconcentration surfaces. MS methods will require preconcentration of ambient $Hg^{II}_{(g)}$ for detection, but the preconcentration surfaces currently available for deployment in the field have limitations which prevent their use in MS methods. Commonly used materials include potassium chloride (KCl)-coated denuders in the Tekran 2537/1130/1135 speciation system (Landis et al., 2002) and membranes deployed in the Reactive Mercury Active System (RMAS) (Luippold et al., 2020b). These membranes include cation exchange membranes (CEM), used for quantitative $Hg^{II}$ measurement; polytetrafluoroethylene membranes (PTFE), used to quantify $Hg^{II}_{(p)}$ (described in Luippold et al., 2020b as PBM); and nylon membranes, used for estimating Hg chemistry.

KCl denuders do not accurately measure $Hg^{II}$ in ambient air due to ozone, humidity, and perhaps other interferences (Lyman et al., 2012; McClure et al., 2013; Huang et al., 2015). PTFE membranes exposed to field conditions have also recently been found to sorb $Hg^{II}_{(g)}$, indicating this membrane type cannot fully separate $Hg^{II}_{(g)}$ and $Hg^{II}_{(p)}$ measurements as intended, and provide $Hg^{II}_{(p)}$ measurements that are biased high (Allen et al., 2024). Although CEM outperform KCl denuders for quantitative $Hg^{II}_{(g)}$ capture (Huang et al., 2013), and are quantitative $Hg^{II}_{(g)}$ sorbants under laboratory conditions (Miller et al., 2019, Dunham-Cheatham 2020), recent work suggests CEM may not be fully quantitative under field conditions. For instance, Dunham-Cheatham et al. (2023) observed lower $Hg^{II}$ concentrations collected on CEM (discussed as RM by Dunham-Cheatham et al. (2023), which includes $Hg^{II}_{(g)}$ and $Hg^{II}_{(p)}$ as defined here) in the field relative to co-located dual-channel system (DCS) $Hg^{II}_{(g)}$ measurements. However, it is possible that the discrepancy between CEM and DCS measurements observed by Dunham-Cheatham et al. (2023) was due to the 2 L $min^{-1}$ flow rate used to collect $Hg^{II}$ on CEM. Use of 2 L $min^{-1}$ flow rates have recently been found to decrease Hg capture efficiency relative to those sampled at a 1 L $min^{-1}$ flow rate (Allen et al., 2024). Increased breakthrough has been detected on downstream CEM during field campaigns compared to CEM exposed to $Hg^{II}_{(g)}$ for short periods in the laboratory (Allen et al., 2024, and this work), and $Hg^{II}$ loss from CEM during long campaign periods has yet to be quantified. CEM are not appropriate for $Hg^{II}$ sample introduction into MS systems, for when heated they generate compounds that interfere with Hg quantification by cold vapor atomic absorption spectroscopy (Gustin et al., 2019), and research done using high $Hg^{II}$ concentrations has suggested that exchange reactions can occur with Hg compounds on the CEM surface (Mao and Khalizov, 2021).

Thermal desorption followed by peak deconvolution of $Hg^{II}$ compounds from nylon membranes deployed in the RMAS is currently the only method available for estimating atmospheric $Hg^{II}$ chemistry (Huang et al., 2013; Luippold et al., 2020a; Gustin et al., 2023). This method compares thermal desorption profiles of unknown $Hg^{II}$ compounds to reference profiles developed from $Hg^{II}$ salts permeated onto nylon membranes to identify potential compound constituents (e.g., -O, -Br/Cl, -N, -S, and –organic Hg compounds). The validity of thermal desorption interpretations depends on how well the desorption behavior of Hg salts represents unknown atmospheric $Hg^{II}$ compounds. Exchange reactions involving $HgBr_2$ and $HgCl_2$ on CEM and nylon membranes have also been observed at above-ambient concentrations (Mao and Khalizov, 2021), suggesting that $Hg^{II}$ compounds desorbed from nylon membranes could be different from atmospheric $Hg^{II}$ compounds initially captured. This may be true of any new $Hg^{II}_{(g)}$ sorptive surface, and should be considered during the validation of new materials.

DCS circumvent the need for $Hg^{II}$ preconcentration by converting $Hg^{II}$ to $Hg^0$ using a thermolyzer and measuring total gaseous Hg. The $Hg^{II}$ concentration can then be calculated by subtracting the $Hg^0$-only fraction of atmospheric air, obtained by scrubbing ambient sample of $Hg^{II}$ with CEM, from the total gaseous Hg measurement. DCS have been successfully calibrated for $Hg^{II}_{(g)}$ measurement in the field, but do not provide $Hg^{II}$ chemistry data (Lyman et al., 2020b).

MS methods that can identify and quantify atmospheric $Hg^{II}$ compounds could be an essential step towards describing Hg chemistry in the atmosphere, but unambiguous determination of the identity of $Hg^{II}$ compounds via MS has not yet been achieved (cf. Deeds et al., 2015; Jones et al. 2016). Given the limitations of current $Hg^{II}$ sorbents, new surfaces that can quantitatively capture $Hg^{II}$ without compound-altering chemistry are needed to preconcentrate ambient samples to levels above MS detection limits. An ideal material for analysis of $Hg^{II}$ compounds by MS will be inert to $Hg^0$, capture and retain all $Hg^{II}$ compounds with high efficiency, not promote compound-altering reactions occurring on the material surface, and release atmospherically representative Hg compounds by thermal desorption for downstream analysis.

Characteristics of promising materials include: ion exchange with a porous or layered crystalline material (Manos and Kanatzidis, 2016); high surface area; and a high melting temperature that would facilitate thermal sample recovery and analysis by MS. Materials functionalized with sulfur, such as thiol, thiosemicarbazide, sulfone, and sulfonamide groups, show promise due to their high affinity for $Hg^{II}$ (Yu et al., 2016). Capture efficiency is increased in base materials by functionalization with active groups that interact with Hg through chemisorption, resulting in the formation of a covalent bond between the Hg atom and material (Ali et al., 2018). However, strong bonding between the material and $Hg^{II}$ may cause the identity of the Hg compound to be lost upon collection. As a result of strong bonding, such a material may not be suitable for subsequent analysis by MS methods. Materials that capture Hg by physisorption processes (electrostatic interactions) may be desirable if Hg compounds do not undergo chemistry on the sorbent surface, as has been observed for CEM and nylon membranes at high concentrations (Mao and Khalizov, 2021).

The objective of this work was to explore chitosan, α-$Al_2O_3$, γ-$Al_2O_3$, poly(1,4-phenylene sulfide), and perfluorosulfonic acid as candidate materials for preconcentration of atmospheric $Hg^{II}$ that would be suitable for

subsequent analysis by MS. A custom-built $Hg^{II}$ permeation calibrator was used to load candidate materials with a known quantity of $Hg^{II}$, for comparison to CEM. $Hg^{II}$ capture by these materials was also compared under field conditions. It was hypothesized that chitosan, $\alpha$-$Al_2O_3$, and $\gamma$-$Al_2O_3$ would quantitatively sorb $Hg^{II}$ under ambient conditions. Chitosan sorbs $Hg^{II}$ through both chelation and electrostatic interactions in liquid matrices via amino and hydroxyl groups (Vieira and Beppu, 2006). $\alpha$-$Al_2O_3$ and $\gamma$-$Al_2O_3$, alumina polymorphs, were potential materials of interest because they are polar compounds, but not acidic, and thus, may attract $Hg^{II}_{(g)}$ compounds without capturing $Hg^0$ (Zheng et al., 2019). Alumina polymorphs are stable at high temperatures (Baronskiy et al., 2022), making them ideal for re-use following thermal desorption. $\alpha$-$Al_2O_3$ and $\gamma$-$Al_2O_3$ differ in thermal stability and specific surface area, and thus may perform differently in terms of capture efficiency and reusability.

## 2 Methods

### 2.1 Materials

CEM, polyethersulfone membranes that are proprietarily treated, were purchased from Pall Corporation (0.8 μm pore size; Mustang-S, P/N MSTGS3R) as sheets and cut to 47 mm diameter discs. PTFE membranes were purchased from Sartorius Stedim Biotech (0.2 μm pore size; P/N1180747). Chitosan (85% deacetylated; P/N J64143.18) and $\alpha$-$Al_2O_3$ (< 1 μm, powder; P/N 0452572.22) were purchased from Thermo Scientific. $\gamma$-$Al_2O_3$ was obtained from Alpha Aesar as a 40 μm powder (P/N 043266.22). $\alpha$-$Al_2O_3$ used in this study had a specific surface area of 2 - 4 $m^2$ $g^{-1}$ and thermal stability of 1200 °C, while the $\gamma$-$Al_2O_3$ had a specific surface area of 100 $m^2$ $g^{-1}$ and thermal stability up to 500 °C. Perfluorosulfonic acid membrane sheets (PFSA-M) were purchased from Sigma Aldrich (trade name Aquivion E98-05, PFSA equivalent weight 980 g $mol^{-1}$ $SO_3H$, 50 μm film thickness; P/N 802697). Poly(1,4-phenylene sulfide) was also purchased from Sigma Aldrich (P/N 182354). Activated carbon was acquired from Aldrich Chemical Company (P/N 292591) as 4-14 mesh granules and crushed for use in experiments as a powder.

A set of glass tubes used to test PFSA-M were sent to SilcoTek for inert coating with deactivated silica (SilcoNert® 2000) to test if this improved $Hg^{II}$ recovery. Based on the results of this test (discussed below), glass tubes without inert coating were used in laboratory and field tests. PTFE frits, which were 10-30 μm in pore size, 2.5 mm thickness, and cut to fit 6.35 mm diameter tubing, were acquired from Savillex (P/N: 730-0065), as were perfluoroalkoxyalkane filter packs used to house membranes (47 mm diameter, P/N 403-21-47-22-21-2). Optima™ HCl (A466-500), KBr (P205-500), $NH_2OH \cdot HCl$ (H330-500), and $SnCl_2$ (T142-500) were obtained from Fisher Scientific. $KBrO_3$ was purchased from Acros Organics (268392500). All reagents were ACS grade or higher and made with 18.2 MΩ-cm type 1 water. Ultra-high purity argon (Linde Gas and Equipment Inc.) was used in sorption tests described in the Appendix. Gas-tight syringes were purchased from Hamilton Company (Reno, Nevada, USA). Flow rates in laboratory tests and field campaigns were controlled with critical flow orifices obtained from Teledyne API (941100).

Candidate materials that could potentially sorb $Hg^{II}_{(g)}$ without capturing $Hg^0$ were identified from the literature and preliminary work was performed to select promising materials for further experimentation. Of the five new materials tested, only three were considered for further investigation. Preliminary work indicated poor recovery of $Hg^{2+}$ from two liquid-spiked poly(1,4-phenylene sulfide) samples when analyzed using a modified EPA Method 1631 digestion (United States Environmental Protection Agency, 2002). This suggested a matrix interference and this material was not tested further. Investigation of PFSA-M was also discontinued after poor performance in $Hg^{II}_{(g)}$ laboratory tests (discussed below). Preliminary work with poly(1,4-phenylene sulfide) is detailed in the Appendix, with additional data for chitosan, $\alpha$-$Al_2O_3$, and $\gamma$-$Al_2O_3$ (Appendix A and B). An alternative digestion method (the appendix to EPA Method 1631) for recovering $Hg^{II}$ from CEM, chitosan, $\alpha$-$Al_2O_3$, and $\gamma$-$Al_2O_3$ was attempted, and these data are also available in the Appendix (Appendix C).

**2.2 Laboratory loading of $Hg^0$ and $Hg^{II}_{(g)}$ onto candidate materials**

Chitosan, $\alpha$-$Al_2O_3$, $\gamma$-$Al_2O_3$, and PFSA-M were tested for quantitative $Hg^0$ and $Hg^{II}_{(g)}$ sorption in the laboratory. To test for sorption of $Hg^0$ to candidate materials, laboratory air was drawn at 1 L min$^{-1}$ through traps containing test material. A syringe was used to inject 1.2 ng $Hg^0$ from a bell jar into the trap (n = 9). As a control, activated carbon was loaded by the same method. Traps containing 30 $\pm$ 5 mg of chitosan, $\alpha$-$Al_2O_3$, $\gamma$-$Al_2O_3$, or shredded PFSA-M were constructed with glass tubing containing a single PTFE frit. The glass tube (6.35 mm internal diameter) was slightly pinched at one end to prevent the frit and test material from being pulled through the trap during loading. Replicates (n = 9) of each material were exposed to laboratory air for three minutes, at the end of which the $Hg^0$ injection was made. Additional traps (n = 9 per material type) that were exposed to laboratory air, but not loaded with $Hg^0$, were used to blank-correct loaded samples. The mass of loaded $Hg^0$ was calculated based on the Dumarey equation (Dumarey et al., 2010). $Hg^0$ recovered from candidate materials was compared both to the calculated mass loaded and to $Hg^0$ recovered from the method control. The syringe ($Hg^0$) tip was placed as close as possible to the materials during loading to minimize loss to the atmosphere or glass tubing.

$Hg^0$ was recovered by combustion (EPA Method 7473) using a direct Hg analyzer (Nippon, MA-3000) (United States Environmental Protection Agency, 2007). This instrument was calibrated ($r^2 \geq 0.999$) with a primary liquid $Hg^{2+}$ standard (Inorganic Ventures, MSHG-1PPM). At the beginning of each analytical run, three aliquots of a secondary standard, National Institute of Standards and Technology Standard Reference Material 1547, were analyzed to demonstrate instrument performance. Two additional aliquots of 1547 were run every 10 samples or fewer to confirm ongoing instrument performance. A recovery within $\pm$ 10% if the certified value was considered acceptable for analysis. This instrument has a detection limit of 0.10 ng $Hg^0$.

Sorption of $Hg^{II}_{(g)}$ to candidate materials was performed with the same procedure, using a custom-built $HgBr_2$ calibrator (Allen et al., 2024; Gačnik et al., 2024) that releases a constant stream of $Hg^{II}_{(g)}$ from a salt-based permeation source. The permeation rate is tightly controlled by maintaining constant temperature, pressure, and He flow over the permeation source. The permeation rate of the calibrator can be determined either gravimetrically or by measurement

with CEM (Lyman et al., 2016; Gačnik et al., 2024). These methods have been demonstrated to be equivalent by Elgiar et al. (2024). The CEM method was used to calculate the permeation rate of the calibrator in this study. Briefly, permeation rates are calculated by exposing CEM to calibrator output, then digesting the CEM by EPA Method 1631. The recovered mass of Hg is then divided by the exposure time to provide a pg s$^{-1}$ permeation rate (blank corrected). Although this calibrator can be used to calibrate Hg$^{II}_{(g)}$ measurements in other systems, it was used here to compare sorptive properties of candidate materials to CEM, by delivering consistent, measurable, quantities of Hg$^{II}_{(g)}$. Chitosan, $\alpha$-Al$_2$O$_3$, $\gamma$-Al$_2$O$_3$, and nylon (n = 9 replicates each), PFSA-M (n = 6 replicates) were exposed to the calibrator output for the same duration, to ensure equal loading of Hg$^{II}_{(g)}$ for each replicate. Triplicate blanks of each material were also exposed to laboratory air for an equivalent period of time. Initial experiments loading CEM in this study measured a $1.76 \pm 0.18$ pg s$^{-1}$ permeation rate (mean $\pm$ standard deviation, n = 9). The permeation rate of this calibrator was reported as $2.2 \pm 0.2$ pg s$^{-1}$ in experiments performed concurrently, discussed elsewhere (Gačnik et al., 2024). The difference in observed permeation rate between these two studies is of significance for the use of this system for calibrating Hg$^{II}$ measurements and should be studied further before it is broadly employed by the research community. A possible explanation for the difference may be the positioning of the calibrator tip at a distance of 2 cm from the CEM during loading (Gačnik et al., 2024) versus at the filter pack inlet (5.5 cm in this work), as HgBr$_2$ is more likely to come in contact with the filterpack when loaded at the inlet. Work by Allen et al. (2024) suggest less than 5% of atmospheric Hg$^{II}$ is sorbed to the PTFE filterpacks after field deployment.

Permeation, as measured by CEM, dropped significantly to $0.42 \pm 0.03$ pg s$^{-1}$ during experiment replication. It was suspected that the chamber containing the permeation tube over heated and shut off, cooling the HgBr$_2$ salt. Returning the heated chamber to 50 °C restored the measured permeation rate to $1.77 \pm 0.06$ (data available in Appendix Table I1). Due to the change in permeation rate across replicate experiments, results are reported as a % of Hg$^{II}_{(g)}$ recovered from CEM, rather than a % of the expected recovery based on the perm rate that was calculated as the mass of Hg$^{II}$ recovered from CEM divided by exposure time (pg s$^{-1}$). Hg$^{II}_{(g)}$ was quantified by following a modified version of EPA Method 1631. Briefly, BrCl solution (1.8% KBr and 1.2% KBrO$_3$ in 32-35% w/w Optima™ HCl) was added to candidate materials and membranes in 1% HCl at a ratio of 3 mL BrCl to 50 mL 1% HCl, and digested at room temperature overnight (see the SI of Dunham-Cheatham et al., 2023 for additional details). Samples were analyzed by cold vapor atomic fluorescence spectroscopy (Tekran 2600-IVS). The instrument was calibrated (r$^2 \geq$ 0.999) at the beginning and end of analysis. A check standard was analyzed every ten samples and the instrument was re-calibrated after a maximum of 30 samples were run. Data were considered acceptable if check standards were recovered within $\pm$ 15% of the true value, as per EPA Method 1631. Blanks for each tested material were collected and analyzed with samples during each experiment, and were used to correct the analyzed value of samples. These blanks were exposed to laboratory air only (no HgBr$_2$) during laboratory HgBr$_2$ exposure tests, or were not exposed to air during field campaigns. Mean recovery on CEM blanks (exposed to laboratory air or not) was $0.03 \pm 0.01$ ng per membrane ($\pm$ 1 standard deviation) across 28 replicates. Mean recovery of $\alpha$-Al$_2$O$_3$, $\gamma$-Al$_2$O$_3$ and chitosan was $0.01 \pm 0.02$, $0.01 \pm 0.01$, and $0.02 \pm 0.01$ ng per target mass (30 mg), respectively (n = 27 for each material).

**2.3 Field comparison of candidate materials and CEM**

Candidate materials were deployed for three one-week sampling campaigns in the summer (late July through mid-September, 2023) at the University of Nevada, Reno College of Agriculture, Biotechnology & Natural Resources Agricultural Experiment Station Valley Road Greenhouse Complex (39.5375, −119.8047, 1370 masl). This sampling location is within 100 m of Interstate-80 and is impacted by vehicle emissions and long-range transport of pollutants (Gustin et al., 2021; Luippold et al., 2020a). Environmental conditions varied between campaigns, with weekly mean temperatures falling from $25.0 \pm 0.5$ °C in the first campaign to $22.0 \pm 3.4$ and $20.8 \pm 0.6$ °C in the last two campaigns. Relative humidity was $20 \pm 4$, $48 \pm 16$, and $35 \pm 5$ % for the first, second, and third campaigns, respectively. Temperature, relative humidity, solar radiation, and precipitation information was obtained from Western Regional Climate Center (https://raws.dri.edu/) measurement station located at the test site (39.53917, -119.806, 1370 masl), and is available in the Appendix (Table D1).

Traps, constructed as described for $Hg^0$ and $Hg^{II}_{(g)}$ sorption tests above, were deployed in inverted RMAS shields (Fig. E1). Three replicates of each material were deployed with a PTFE membrane upstream (to separate $Hg^{II}_{(g)}$ and $Hg^{II}_{(p)}$), and three replicate traps were deployed without an upstream PTFE membrane (providing a total $Hg^{II}$ measurement). A CEM was deployed behind each candidate material to capture breakthrough $Hg^{II}_{(g)}$. Filter packs containing two consecutive CEM, both with and without an upstream PTFE membrane (n = 3 for each configuration), were co-deployed with candidate materials. Critical flow orifices controlled the flow across all collection materials at 1 L min$^{-1}$, and the flow rate through each trap or filter pack assembly was measured as standard flow at the beginning and end of each campaign using a volumetric air flow calibrator (BGI tetraCal). Measured masses of $Hg^{II}_{(g)}$ in each trap or filter pack were divided by the volume of air sampled, calculated as the average of flow rates measured at the beginning and end of deployment, in L min$^{-1}$, multiplied by the total sampling time in min, to calculate a $Hg^{II}_{(g)}$ concentration sampled in ambient air. All membranes and candidate materials were digested by the modified EPA Method 1631 procedure described above.

**2.4 Statistical analysis**

One-way ANOVA and Tukey's honestly significant difference tests ($\alpha \leq 0.05$), comparing the recovery of $Hg^{II}$ between material types, were performed using R (R Core Team, 2023, version 2023.06.2+561). Means, standard deviations, and t-tests were calculated using Excel 2016.

**3 Results and discussion**

**3.1 Laboratory tests for $Hg^0$ and $Hg^{II}_{(g)}$ sorption**

In this work, chitosan, $\alpha$-Al$_2$O$_3$, and $\gamma$-Al$_2$O$_3$, were tested for sorption of $Hg^0$ and $Hg^{II}$ in the laboratory and the field. Additionally, preliminary $Hg^{II}_{(g)}$ sorption tests with poly (1,4-phenylene sulfide) and PFSA-M were performed, but these materials were abandoned when a matrix interference was identified for (1,4-phenylene sulfide) and PFSA-M demonstrated poor $Hg^{II}_{(g)}$ recovery. A summary of the materials tested and outcomes are available in Table 1, and the details of preliminary work are available in Appendix A and B. No quantifiable $Hg^0$ was recovered

from blanks or $Hg^0$-loaded chitosan, $\alpha$-$Al_2O_3$, nor $\gamma$-$Al_2O_3$, except for one $Hg^0$-loaded $\gamma$-$Al_2O_3$ trap that had low recovery (0.11 ng). $Hg^0$ recovery from activated carbon was $1.3 \pm 0.4$ ng (mean $\pm$ standard deviation), indicating reasonably good agreement with the expected recovery of 1.1 ng, based on the Dumarey equation (Fig. F1).

Table 1. Outcomes of materials tested.

| Material tested | Outcome of findings |
|---|---|
| Chitosan | $Hg^{II}_{(g)}$ was recovered quantitatively from chitosan under laboratory conditions, but less $Hg^{II}_{(g)}$ was recovered compared to CEM during field deployments. Ambient humidity may have interfered with $Hg^{II}_{(g)}$ capture by chitosan. |
| $\alpha$-$Al_2O_3$ | Less $Hg^{II}_{(g)}$ was recovered from $\alpha$-$Al_2O_3$ compared to CEM under both laboratory and field conditions. EPA Method 1631 may be insufficient to quantitatively recover $Hg^{II}_{(g)}$ from this matrix. |
| $\gamma$-$Al_2O_3$ | Less $Hg^{II}_{(g)}$ was recovered from $\gamma$-$Al_2O_3$ compared to CEM under both laboratory and field conditions. EPA Method 1631 may be insufficient to quantitatively recover $Hg^{II}_{(g)}$ from this matrix. |
| Poly(1,4-phenylene sulfide) | Poor recovery of liquid $Hg^{II}$ from spiked poly(1,4-phenylene sulfide) indicated a matrix interference when digested by EPA Method 1631. |
| Polyflourosulfonic acid membrane (PFSA-M) | Poor recovery of $Hg^{II}_{(g)}$ from this material was observed compared to CEM under laboratory conditions. |

Chitosan traps recovered $99 \pm 36\%$ of the loaded $Hg^{II}_{(g)}$ compared to the CEM, while $\alpha$-$Al_2O_3$, $\gamma$-$Al_2O_3$, nylon, and PFSA-M recovered less ($86 \pm 15\%$, $69 \pm 21\%$, $81 \pm 7\%$, and $26 \pm 7\%$, respectively) (Fig. 1). Due to the smaller quantities of $Hg^{II}_{(g)}$ loaded following the drop in permeation, small variation in mass $Hg^{II}_{(g)}$ recovered

resulted in a larger % variation. Variation in mass $Hg^{II}_{(g)}$ recovered from both the candidate material and CEM also contributed. Given the low recovery of $Hg^{II}_{(g)}$ from PFSA-M, it was not tested further. As breakthrough was not quantified for this material, it is unclear if the low recovery is due to low capture efficiency of the material itself, or if the geometry of the trap (shredded membrane packed into glass tubing) permitted greater breakthrough. If this membrane material could be made porous, or functionalized onto a porous substrate, it may demonstrate increased

capture of $Hg^{II}_{(g)}$ due to the presence of sulfonic acid functional groups. A comparison of $Hg^{II}_{(g)}$ recoveries was made between PFSA-M membranes that were loaded with $Hg^{II}_{(g)}$ in glass tubing coated with deactivated fused silica, or in uncoated glass tubing. Although it was hypothesized that the coating would reduce $Hg^{II}_{(g)}$ sorption to the glass tubing (Jones et al., 2016), $Hg^{II}_{(g)}$ recovery was not statistically different between PFSA-M samples loaded in deactivated fused silica coated or uncoated tubes ($p > 0.05$; Fig. G1). $Hg^{II}_{(g)}$ recovered on CEM downstream of other

candidate materials provided a measurement of $Hg^{II}_{(g)}$ breakthrough, calculated as a % of the sum of $Hg^{II}_{(g)}$ recovered from the candidate material plus the CEM (n = 6 replicates for chitosan, $\alpha$-$Al_2O_3$, $\gamma$-$Al_2O_3$, and nylon). An average of $\leq 5\%$ $Hg^{II}_{(g)}$ was recovered downstream of candidate materials, and no quantifiable $Hg^{II}_{(g)}$ was recovered on the second-in-line CEM behind a CEM.

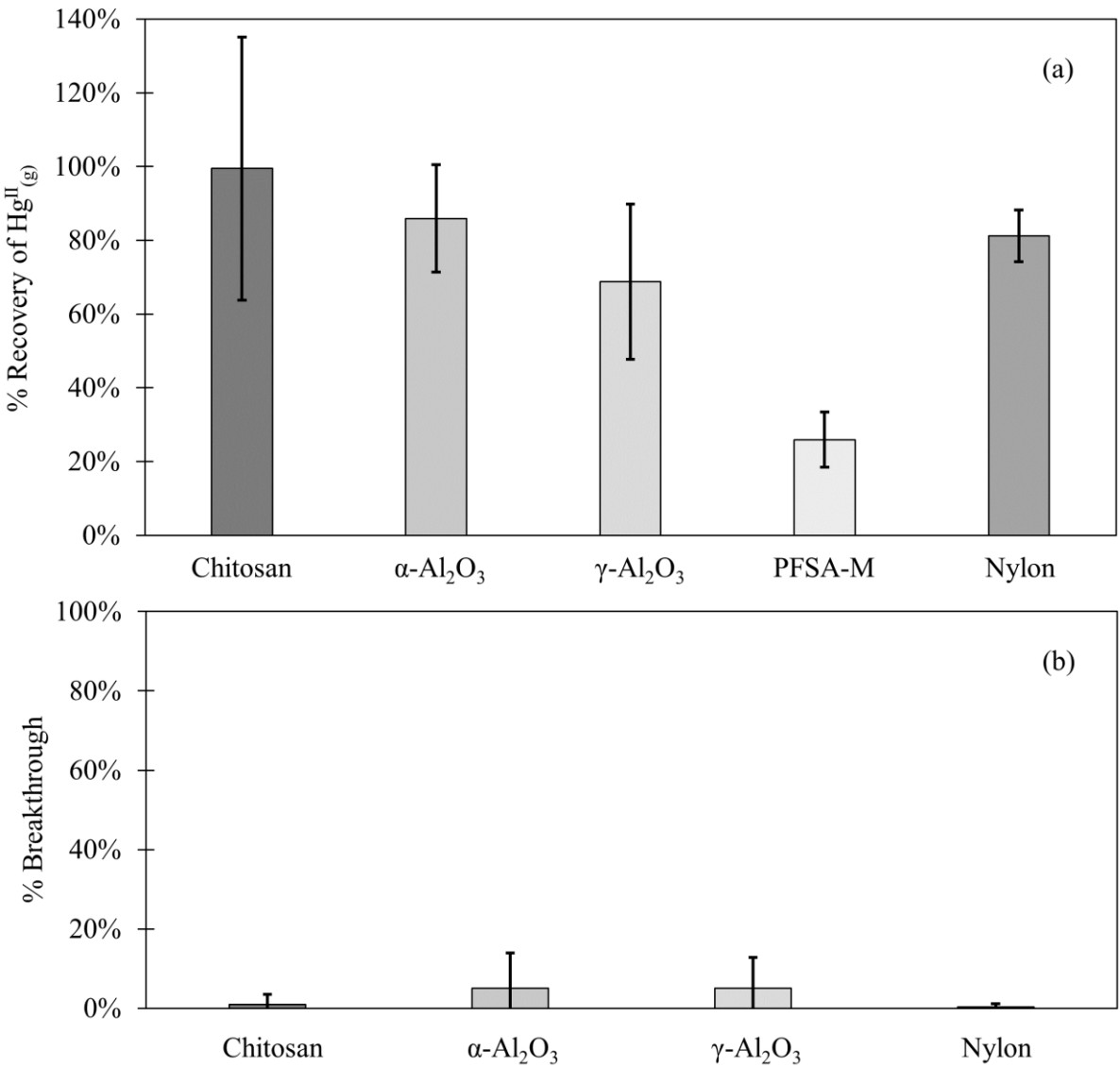

Figure 1. (a) $Hg^{II}_{(g)}$ recovered from candidate materials (n = 9 each for chitosan, α-Al$_2$O$_3$, γ-Al$_2$O$_3$, and nylon, and n = 6 for PFSA-M) loaded with $HgBr_2$ in laboratory air, as a percentage of $HgBr_2$ recovered on CEM. (b) $Hg^{II}_{(g)}$ breakthrough from candidate materials (n = 6 each) as a % of $HgBr_2$ collected on CEM. Error bars represent one standard deviation from the mean.

The relatively low recovery of $Hg^{II}_{(g)}$ from α-Al$_2$O$_3$ and γ-Al$_2$O$_3$, as well as the minimal breakthrough, indicated that $Hg^{II}_{(g)}$ was either lost during loading, possibly to the glass tubing, or it was not recovered from the material matrix by BrCl digestion. Al$_2$O$_3$ has been previously used to capture and thermally reduce $Hg^{II}$ to $Hg^0$ with high efficiency in an inert atmosphere (Gačnik et al., 2022), suggesting the EPA Method 1631 digestion is not sufficient to recover $Hg^{II}$ sample from this matrix. Chitosan, an organic compound, is more easily decomposed, and thus, sorbed Hg is made more available for analysis by acid digestion compared to oxide crystals like α-Al$_2$O$_3$ and γ-Al$_2$O$_3$. This may explain why a higher % recovery was observed from chitosan compared to α-Al$_2$O$_3$ and γ-Al$_2$O$_3$, and

highlights the need to consider alternative digestion methods, or possibly thermal desorption, and utilize matrix-matched certified reference materials when considering new surfaces for quantitative $Hg^{II}_{(g)}$ capture.

Digestions with HF and $HNO_3$ have been used to recover metals from refractory silicates and oxides (Zimmermann et al., 2020), and EPA Method 3052 (United States Environmental Protection Agency, 1996) is an established method for Hg. It aims to completely decompose and dissolve the sample by microwave digestion with HF and $HNO_3$, but also offers alternative matrix-specific reagent mixtures with HCl and $H_2O_2$. This method notes that it may not be suitable for some oxides, including $Al_2O_3$ and $TiO_2$, among others, and that target analytes (including Hg) can be sequestered by undecomposed sample, leading to low recovery. Re-adsorption of Hg by residual sample matrix during digestion is also noted for activated carbon matrices in the appendix to EPA Method 1631 and could explain low $Hg^{II}_{(g)}$ recovery from the $\alpha$-$Al_2O_3$ and $\gamma$-$Al_2O_3$ in this study. Microwave digestion of $Hg^{II}_{(g)}$ loaded carbon and $\gamma$-$Al_2O_3$ with $HBF_4$ was attempted as a safer alternative to digestion with HF (Zimmermann et al., 2020), but high background Hg in analytical-grade reagents made data inconclusive. Direct Hg analyzers conveniently overcome matrix interferences by combusting the sample, but atmospheric samples will need to be preconcentrated over 2-week campaigns to collect enough Hg to exceed analytical detection limits, limiting the utility of this method for analyzing $Hg^{II}$ trends over short timescales.

**3.2 $Hg^{II}$ recovery from candidate materials in the field**

Candidate materials were deployed with or without a PTFE membrane upstream, and with a CEM downstream. Of the total $Hg^{II}$ recovered from PTFE + $Al_2O_3$ traps, 66% was recovered from the PTFE portion of $\alpha$-$Al_2O_3$ traps and 55% of $Hg^{II}$ was recovered from the PTFE on $\gamma$-$Al_2O_3$ traps, indicating that half or more of the $Hg^{II}$ recovered was particulate-bound or $Hg^{II}_{(g)}$ sorbed to particles (see below). More $Hg^{II}_{(g)}$ was captured on chitosan behind PTFE compared to the equivalent $\alpha$-$Al_2O_3$ and $\gamma$-$Al_2O_3$ traps (an average across all three campaigns of $7 \pm 3$ pg m$^{-3}$ $\alpha$-$Al_2O_3$ vs $28 \pm 43$ pg m$^{-3}$ $\gamma$-$Al_2O_3$ and $46 \pm 36$ pg m$^{-3}$ chitosan), but not as much as compared to CEM ($91 \pm 45$ pg m$^{-3}$). This agrees with the laboratory tests that show poor $Hg^{II}_{(g)}$ recovery from $\alpha$-$Al_2O_3$ and $\gamma$-$Al_2O_3$ and relatively greater $Hg^{II}_{(g)}$ recovery from chitosan. Total $Hg^{II}$ recovery from the entire trap assembly (PTFE membrane (if present) + candidate material or first-in-line CEM + breakthrough CEM) was not statistically different between chitosan and CEM traps during any campaign. Recovery was statistically lower for $\alpha$-$Al_2O_3$ and $\gamma$-$Al_2O_3$ in the first and third campaigns, and higher for $\alpha$-$Al_2O_3$ during the second campaign (Fig. H1(a)). Field measurements included a downstream CEM that captured $Hg^{II}$ not sorbed by candidate materials (i.e. "breakthrough"), if present. These data also suggest that either BrCl digestion was not sufficient to recover $Hg^{II}$ from $Al_2O_3$ matrices or reduction was occurring on the material surface during the week-long sampling period and $Hg^{II}$ was lost as $Hg^0$.

Recent work by Allen et al. (2024) demonstrated that $Hg^{II}_{(g)}$ can be sorbed by particulates on PTFE filters, suggesting that $Hg^{II}_{(p)}$ measurements are biased high with additional $Hg^{II}_{(g)}$ sorbed. For this reason, $Hg^{II}$ recovered on PTFE filters was added to $Hg^{II}_{(g)}$ recovered from downstream candidate materials to yield a total $Hg^{II}$ measurement, and these data were combined with $Hg^{II}$ measurements from candidate materials without upstream PTFE. The sum of $Hg^{II}_{(p)} + Hg^{II}_{(g)}$ recovered from PTFE + CEM, respectively, has been well correlated with $Hg^{II}$ measurements on CEM in previous work (Gustin et al., 2019; Gustin et al., 2023). CEM (n = 6, three of which included $Hg^{II}$ recovery from

PTFE + CEM) recovered the highest mass $Hg^{II}$ m$^{-3}$ air sampled of all materials tested (Fig. 2(a)). More than 20% of the $Hg^{II}$ recovered from the entire trap (PTFE membrane, candidate material, and breakthrough CEM) was recovered downstream of candidate materials (Fig. 2(b)), indicating that chitosan, $\alpha$-Al$_2$O$_3$, and $\gamma$-Al$_2$O$_3$ did not quantitatively

measure $Hg^{II}$ under field conditions. There was a decrease in the $Hg^{II}$ measured by all materials in the second and third campaigns that coincided with periods of rain and increased humidity, which is consistent with observations of $Hg^{II}$ washout during rain events (Kaulfus et al., 2017). Of the candidate materials, chitosan performed the best during the first campaign, recovering a similar quantity of $Hg^{II}$ as CEM, but decreased in relative recovery during the second and third campaigns. Chitosan is highly hygroscopic (Szymańska and Winnicka, 2015), and the amino functional

groups on chitosan are easily protonated at pH $< 6$, thus it is possible the increased humidity led to a decrease in $Hg^{II}_{(g)}$ sorption capacity due to electrostatic repulsion between protonated amino groups and $Hg^{II}_{(g)}$ (Vieira and Beppu, 2006).

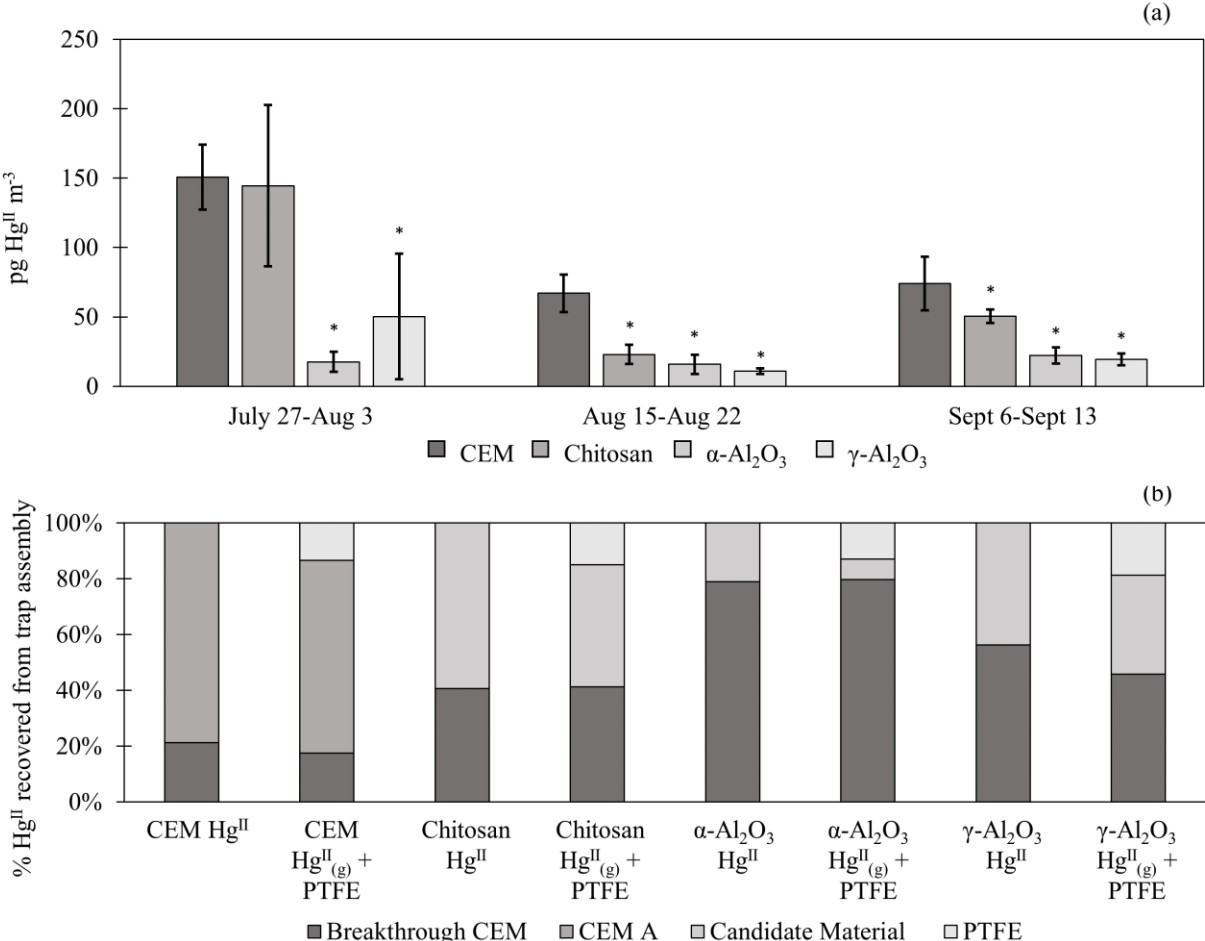

Figure 2. (a) $Hg^{II}$ recovery from PTFE + candidate materials deployed in the field for three one-week campaigns. $Hg^{II}$

recovery from traps with first-in-line PTFE membranes ($Hg^{II}_{(g)}$ +$Hg^{II}_{(p)}$) were combined with traps without PTFE ($Hg^{II}$) for statistical analysis (n = 6 per campaign). Data do not include $Hg^{II}_{(g)}$ recovered on breakthrough CEM. Error

bars are one standard deviation from the mean. Asterisks (*) indicate a statistically different recovery of $Hg^{II}_{(g)}$ on candidate materials compared to CEM (ANOVA, $\alpha \leq 0.05$). (b) The % of $Hg^{II}$ recovered from each portion of the trap assembly, including from the PTFE, candidate material, first-in-line CEM (CEM A), and/or breakthrough CEM. $Hg^{II}$ traps with no upstream PTFE are shown separately from $Hg^{II}_{(g)}$ + PTFE traps.

## 4 Conclusions

CEM outperformed chitosan, $\alpha$-$Al_2O_3$, and $\gamma$-$Al_2O_3$ for $Hg^{II}_{(g)}$ measurement in the laboratory and the field, indicating they do not quantitatively capture $Hg^{II}$. Candidate materials did not collect $Hg^0$. Low recoveries of $Hg^{II}_{(g)}$ from $\alpha$-$Al_2O_3$ and $\gamma$-$Al_2O_3$ may be due to insufficient digestion methods, demonstrating a need for using matrix-specific methods with certified reference materials when testing alternative materials in the future. Promising materials should be tested for: sorption of $Hg^{II}_{(g)}$ and $Hg^0$ capture efficiency for a broad range of representative Hg compounds (Dunham-Cheatham et al., 2020); the potential for chemical transformation on the material surface; potential reactions between the Hg sample and other atmospheric constituents, including interferences with humidity (Huang and Gustin, 2015) and ozone (McClure et al., 2014); and for performance under both laboratory and field conditions.

## Appendices

## Appendix A: Preliminary assessment of gaseous elemental mercury sorption in an argon atmosphere to poly(1,4-phenylene sulfide), chitosan, perfluorosulfonic acid, and $\alpha$-$Al_2O_3$

Cold vapor atomic fluorescence spectroscopy was used to characterize the loss of Hg in an argon (Ar) carrier gas following injection of gaseous elemental mercury ($Hg^0$) through a trap containing poly(1,4-phenylene sulfide) (PPS), chitosan, or a shredded perfluorosulfonic acid membrane (PFSA-M). The testing apparatus (Fig. A1) consisted of: an Ar cylinder that provided carrier gas and pressure to the sample line; a PTFE sample line into which a trap containing a test material could be inserted with an upstream $Hg^0$ injection port; a thermolyzer ($> 650$ °C) to convert gaseous oxidized mercury ($Hg^{II}_{(g)}$) to $Hg^0$; a gold cartridge to collect $Hg^0$ for analysis; and a Tekran 2500 to measure $Hg^0$. Materials were loaded with a gas-tight syringe by injecting $Hg^0$ from a temperature-stabilized source through the injection port upstream of a trap containing a sorbent material. The quantity of $Hg^0$ loaded was calculated based on the Dumarey equation (Dumarey et al., 2010). Sorbent traps containing 29.4 mg PPS, 29.4 mg chitosan, or a half of a 47 mm diameter PFSA-M (one replicate each) were constructed as described in the main text with 6.35 mm internal diameter uncoated glass tubing and quartz wool plugs.

To test for sorption of $Hg^0$ to PPS, chitosan, and PFSA-M, the peak area of 0.4 ng $Hg^0$ detected downstream of traps containing material was compared to a baseline peak area of 0.4 ng $Hg^0$ injected through a glass trap without sorbent material. Peaks observed following an injection indicated no or partial sorption of $Hg^0$ to the candidate material. A student's t-test ($\alpha = 0.05$) was used to assess if peak area of $Hg^0$ detected downstream of the candidate material (n = 5 injections of 0.4 ng $Hg^0$ each, relative standard deviation (RSD) < 10% for PPS) was statistically

different from $Hg^0$ peak area detected downstream of the empty trap (n = 5 injections, RSD = 7%). The absence of a peak following an injection indicated complete sorption of $Hg^0$ to the trap, while significantly lower peak areas indicated partial sorption, and peak areas equivalent to peaks detected downstream of an empty trap indicated no $Hg^0$ sorption to the candidate material. PPS and chitosan did not sorb any $Hg^0$. More $Hg^0$ was recovered downstream of PFSA-M membranes compared to empty glass traps (RSD of injections < 5 %, p < 0.01). The reason for this was unclear and this test was repeated a second time with the same outcome (a total of 20 injections through each empty and PFSA-M trap over two days). It was concluded that PFSA-M did not sorb $Hg^0$.

A trap containing 14 mg $\alpha$-$Al_2O_3$ was tested using a similar procedure with a few minor differences. The glass tubing was pinched at one end and contained PTFE frits, rather than quartz wool plugs, to prevent material movement, and the Tekran 2500 was calibrated with $Hg^0$ using the bell jar method ($r^2 > 0.998$) to quantify sorption by $\alpha$-$Al_2O_3$. The quantity of $Hg^0$ injected was also increased from 0.4 to 1.1 ng Hg so analysis was performed on a Hg mass in the middle of the calibration range. $\alpha$-$Al_2O_3$ sorbed 40% of the injected $Hg^0$, demonstrating that significant sorption of $Hg^0$ is possible in an Ar atmosphere.

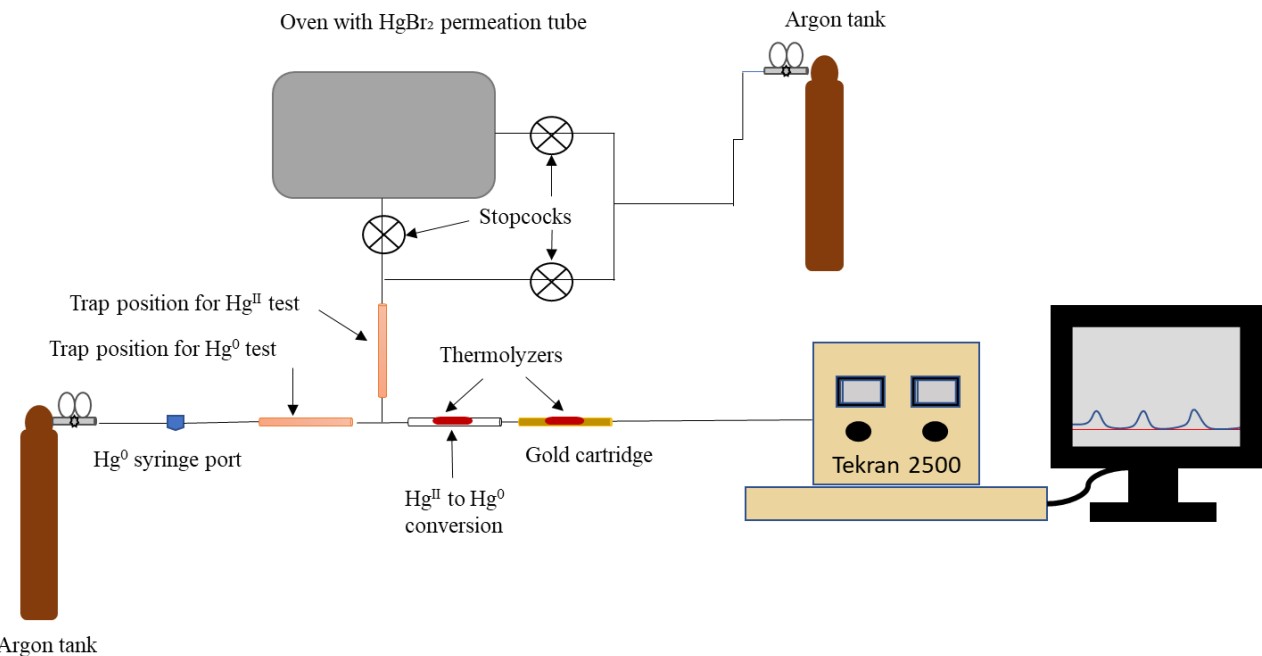

**Fig. A1:** Experimental setup for $Hg^0$ and $Hg^{II}_{(g)}$ sorption to candidate materials in an argon atmosphere. During $Hg^0$ sorption testing, the line containing the $Hg^{II}_{(g)}$ permeation tube was disconnected. Lines leading into the permeation source and to the Tekran 2500 were heated to a nominal 83 °C with heat tape and insulated with aluminum foil to encourage $Hg^{II}_{(g)}$ movement through the system and reduce photo-effects. Components in figure are not to scale.

**Appendix B: Preliminary assessment of $Hg^{II}_{(g)}$ sorption to PFSA-M and PPS in an argon atmosphere**

To test $Hg^{II}_{(g)}$ sorption to candidate materials, the experimental apparatus included a heated (nominally 90 °C) impinger containing a $HgBr_2$ permeation tube (Fig. A1) that could provide controlled injections of $Hg^{II}_{(g)}$ through a candidate material trap (35.6 mg PPS or a half of a 47 mm diameter PFSA-M membrane). For $Hg^{II}$ tests, the sample line was also kept warm with heat tape (~83 °C) and insulated with aluminum foil. The same injection procedure as described above was used to determine $Hg^{II}_{(g)}$ sorption to PPS and PFSA-M, although the empty trap contained PTFE plugs rather than quartz wool. To confirm $Hg^{II}_{(g)}$ source stability over time, peak area was observed through an empty trap before and after $Hg^{II}_{(g)}$ injections, and a t-test was used to check that peak area from injections did not differ from the beginning of the test to the end of the test. Beginning and end injections were not statistically different in peak area.

Statistically different peak areas ($p < 0.05$) were observed between an empty glass trap and both PFSA-M and PPS, indicating these materials sorbed $Hg^{II}_{(g)}$ (n = 5, RSD $\leq$ 10% for injections through an empty trap; n = 5, RSD < 10% for PFSA-M; and n = 5, RSD = 14% for injections through the PPS trap).

**Appendix C: Appendix to EPA Method 1631**

An alternative digestion method was attempted to improve recovery of $Hg^{II}_{(g)}$ from $\alpha$-$Al_2O_3$ and $\gamma$-$Al_2O_3$. The appendix to EPA Method 1631 is a similar digestion procedure to Method 1631, but with an additional leaching step using aqua regia (3:1 $HCl$:$HNO_3$) before digestion with BrCl. A $Hg^{II}_{(g)}$ calibrator (described in the main text) was used to load materials with a known mass of $Hg^{II}_{(g)}$, in lieu of an appropriate certified reference material. An expected 0.25 ng $Hg^{II}_{(g)}$ was loaded onto each material (30 $\pm$ 5 mg chitosan, $\alpha$-$Al_2O_3$, or $\gamma$-$Al_2O_3$; n = 3 traps each), which was then digested by the appendix to EPA Method 1631 and analyzed by cold vapor atomic fluorescence. Activated carbon (30 $\pm$ 5 mg) was used downstream of chitosan, $\alpha$-$Al_2O_3$, and $\gamma$-$Al_2O_3$ to measure $Hg^{II}_{(g)}$ not captured by the candidate material. A second-in-line cation exchange membrane (CEM) captured breakthrough from a first-in-line CEM. All measurements were blank corrected with the appropriate material. Results were highly variable for $\alpha$-$Al_2O_3$ (0.13 $\pm$ 0.12 ng $Hg^{II}_{(g)}$ recovered on $\alpha$-$Al_2O_3$, and 0.17 $\pm$ 0.15 on breakthrough carbon), and no $Hg^{II}_{(g)}$ was recovered from $\gamma$-$Al_2O_3$ with little $Hg^{II}_{(g)}$ recovered from downstream carbon (0.03 $\pm$ 0.06 ng $Hg^{II}_{(g)}$). CEM reasonably recovered the expected loaded mass (0.23 $\pm$ 0.06 ng $Hg^{II}_{(g)}$), with no quantifiable breakthrough (Fig. C1). The expected 0.25 ng of $Hg^{II}_{(g)}$ was reasonably recovered from traps containing chitosan and breakthrough activated carbon (0.13 $\pm$ 0.06 ng recovered on chitosan, 0.1 ng recovered on downstream activated carbon) indicating that the appendix method worked to recover the mass balance from CEM, chitosan, and carbon matrices. The lack of $Hg^{II}_{(g)}$ recovery from $\alpha$-$Al_2O_3$, $\gamma$-$Al_2O_3$ and downstream carbon suggest that $\alpha$-$Al_2O_3$ and $\gamma$-$Al_2O_3$ may be sorbing $Hg^{II}$ but this digestion method is insufficient to quantify it. The appendix to EPA Method 1635 was chosen as a digestion procedure because it is intended for recalcitrant matrices, including coal; however, aqua regia has a matrix-dependent leaching efficiency (Zimmermann et al., 2020). A certified reference material matrix matched to $Al_2O_3$ may conclusively demonstrate this.

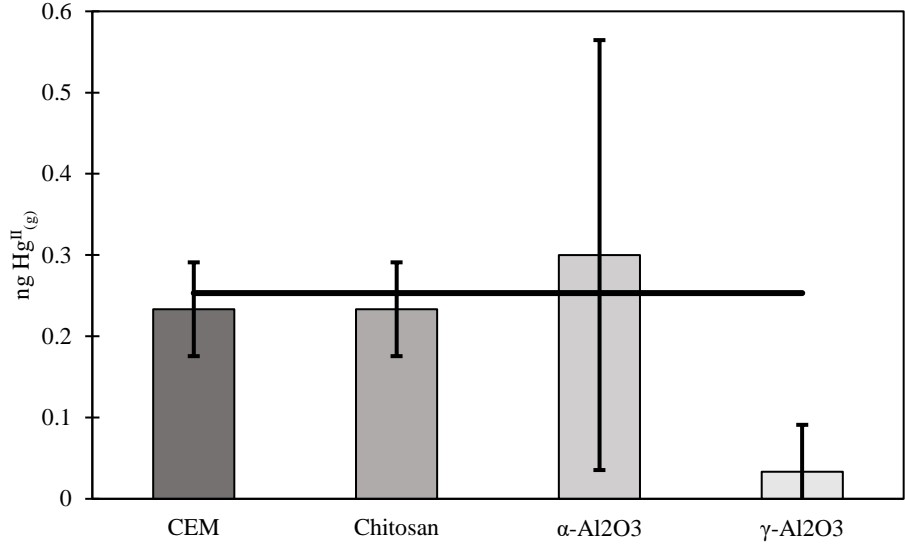

**Fig. C1:** $Hg^{II}_{(g)}$ (ng) recovered by the appendix to EPA Method 1631 on chitosan, $\alpha$-Al$_2$O$_3$, and $\gamma$-Al$_2$O$_3$ compared to an equivalent mass of $Hg^{II}_{(g)}$ loaded on CEM. Masses include $Hg^{II}_{(g)}$ measured on downstream activated carbon. Error bars represent one standard deviation from the mean.

**Table D1:** Environmental conditions during Reactive Mercury Active System (RMAS) campaigns. These data were downloaded as shown from the Western Regional Climate Center (https://raws.dri.edu/). The measurement station was located at the test site (39.53917, -119.806, 1370 masl). Means and standard deviations (highlighted in green) were calculated by the authors from the presented data.

| Date | Total Solar Radiation (ly) | Mean Air Temperature (°C) | Max. Mean Air Temperature (°C) | Min. Mean Air Temperature (°C) | Mean Relative Humidity (%) | Max. Mean Relative Humidity (%) | Min. Mean Relative Humidity (%) | Total Precipitation (cm) |
|---|---|---|---|---|---|---|---|---|
| **Campaign 1** | | | | | | | | |
| 7/27/23 | 759.1 | 24.9 | 33.0 | 14.4 | 28 | 56 | 15 | 0.0 |
| 7/28/23 | 761.8 | 25.6 | 33.2 | 13.7 | 23 | 50 | 13 | 0.0 |
| 7/29/23 | 554.3 | 25.1 | 33.4 | 17.4 | 24 | 46 | 13 | 0.0 |
| 7/30/23 | 766.2 | 25.6 | 33.7 | 13.7 | 23 | 50 | 11 | 0.0 |
| 7/31/23 | 710.7 | 25.4 | 34.2 | 14.2 | 23 | 50 | 12 | 0.0 |
| 8/1/23 | 681.4 | 24.7 | 35.1 | 15.3 | 28 | 49 | 13 | 0.0 |
| 8/2/23 | 612.0 | 24.8 | 34.4 | 15.7 | 33 | 59 | 15 | 0.0 |
| 8/3/23 | 713.6 | 24.3 | 32.8 | 14.9 | 32 | 64 | 17 | 0.0 |
| Weekly mean | 694.9 | 25.0 | 33.7 | 14.9 | 27 | 53 | 14 | 0.0 |
| Standard deviation | 76.6 | 0.5 | 0.8 | 1.3 | 4 | 6 | 2 | 0.0 |
| **Campaign 2** | | | | | | | | |
| 8/15/23 | 545.6 | 23.7 | 34.4 | 13.1 | 50 | 96 | 18 | 0.0 |
| 8/16/23 | 598.7 | 25.5 | 36.7 | 14.9 | 41 | 72 | 16 | 0.0 |
| 8/17/23 | 527.9 | 25.7 | 33.7 | 19.3 | 44 | 77 | 21 | 0.0 |
| 8/18/23 | 547.7 | 23.7 | 33.2 | 14.4 | 42 | 76 | 18 | 0.0 |
| 8/19/23 | 528.4 | 23.3 | 32.9 | 15.2 | 41 | 71 | 15 | 0.0 |
| 8/20/23 | 215.7 | 18.0 | 21.6 | 15.0 | 76 | 98 | 53 | 0.5 |
| 8/21/23 | 317.9 | 17.8 | 23.6 | 13.7 | 73 | 98 | 44 | 0.3 |
| 8/22/23 | 414.5 | 18.1 | 25.5 | 12.6 | 66 | 96 | 34 | 0.1 |
| Weekly mean | 462.1 | 22.0 | 30.2 | 14.8 | 54 | 86 | 27 | 0.1 |
| Standard deviation | 133.9 | 3.4 | 5.7 | 2.1 | 15 | 12 | 15 | 0.2 |
| **Campaign 3** | | | | | | | | |

| | | | | | | | |
|---|---|---|---|---|---|---|---|
| 9/6/23 | 587.7 | 20.4 | 28.8 | 11.8 | 53 | 91 | 24 | 0.0 |
| 9/7/23 | 595.8 | 20.1 | 30.3 | 9.9 | 47 | 87 | 15 | 0.0 |
| 9/8/23 | 597.3 | 20.0 | 30.4 | 8.3 | 42 | 81 | 12 | 0.0 |
| 9/9/23 | 570.2 | 21.1 | 31.3 | 10.5 | 45 | 79 | 17 | 0.0 |
| 9/10/23 | 580.9 | 21.4 | 31.0 | 11.0 | 44 | 82 | 20 | 0.0 |
| 9/11/23 | 565.2 | 21.7 | 31.7 | 11.2 | 40 | 76 | 18 | 0.0 |
| 9/12/23 | 523.0 | 21.1 | 30.5 | 10.8 | 39 | 68 | 19 | 0.0 |
| 9/13/23 | 551.8 | 20.6 | 28.9 | 11.5 | 40 | 70 | 21 | 0.0 |
| Weekly mean | 571.5 | 20.8 | 30.4 | 10.6 | 44 | 79 | 18 | 0.0 |
| Standard deviation | 25.0 | 0.6 | 1.1 | 1.1 | 5 | 8 | 4 | 0.0 |

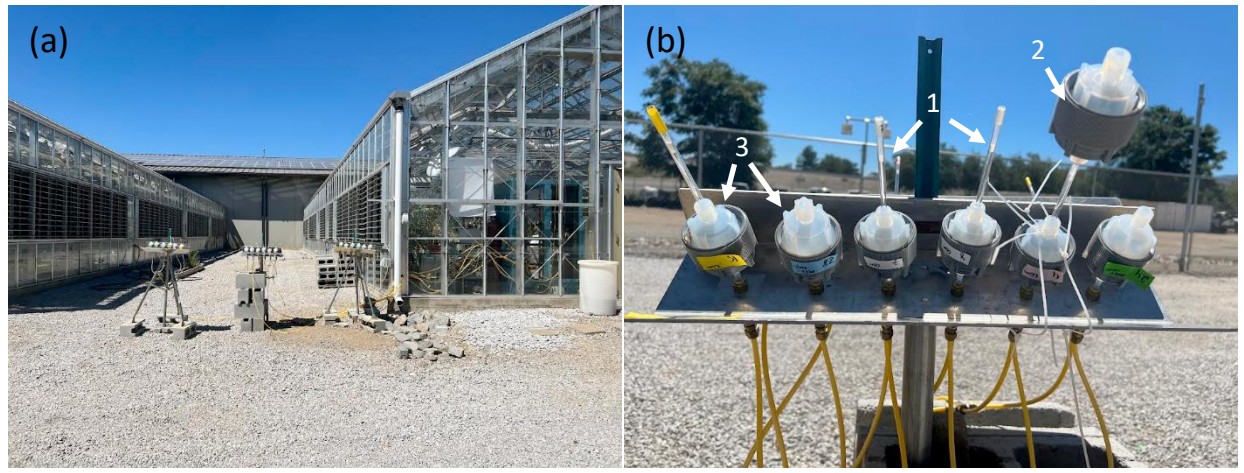

**Fig. E1:** (a) Field campaign of candidate materials and CEM in inverted RMAS shields. (b) A close-up of an inverted RMAS shield holding: (1) glass traps with candidate materials; (2) PTFE membranes; and (3) breakthrough CEM.

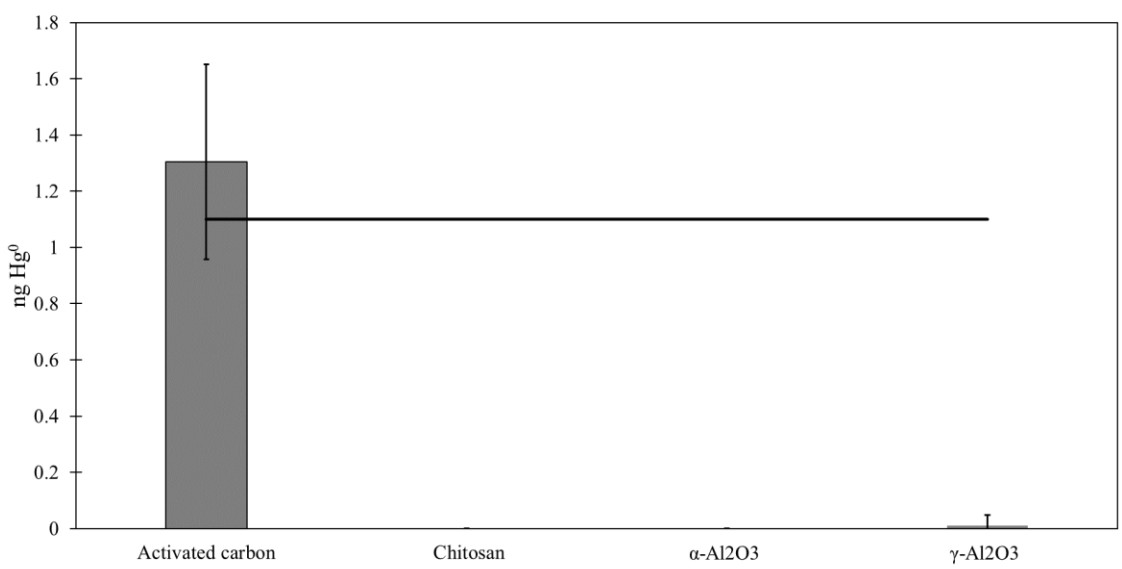

**Fig. F1:** Recovery of $Hg^0$ from activated carbon and candidate materials loaded in ambient air by syringe injection. The black line indicates the calculated mass (1.1 ng $Hg^0$) loaded based on the Dumarey equation.

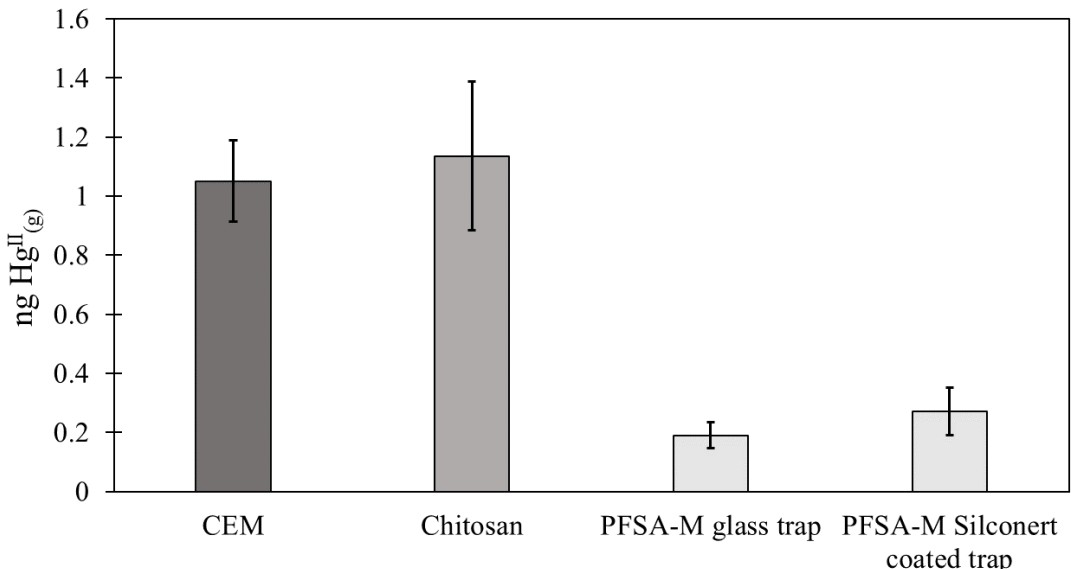

**Fig. G1:** $Hg^{II}_{(g)}$ recovery from PFSA-M using deactivated fused silica coated and uncoated glass tubes, compared to recovery from CEM and chitosan. The recovery of $Hg^{II}_{(g)}$ from PFSA-M was not statistically different between coated and uncoated glass traps ($p > 0.05$, two-sample t-test).

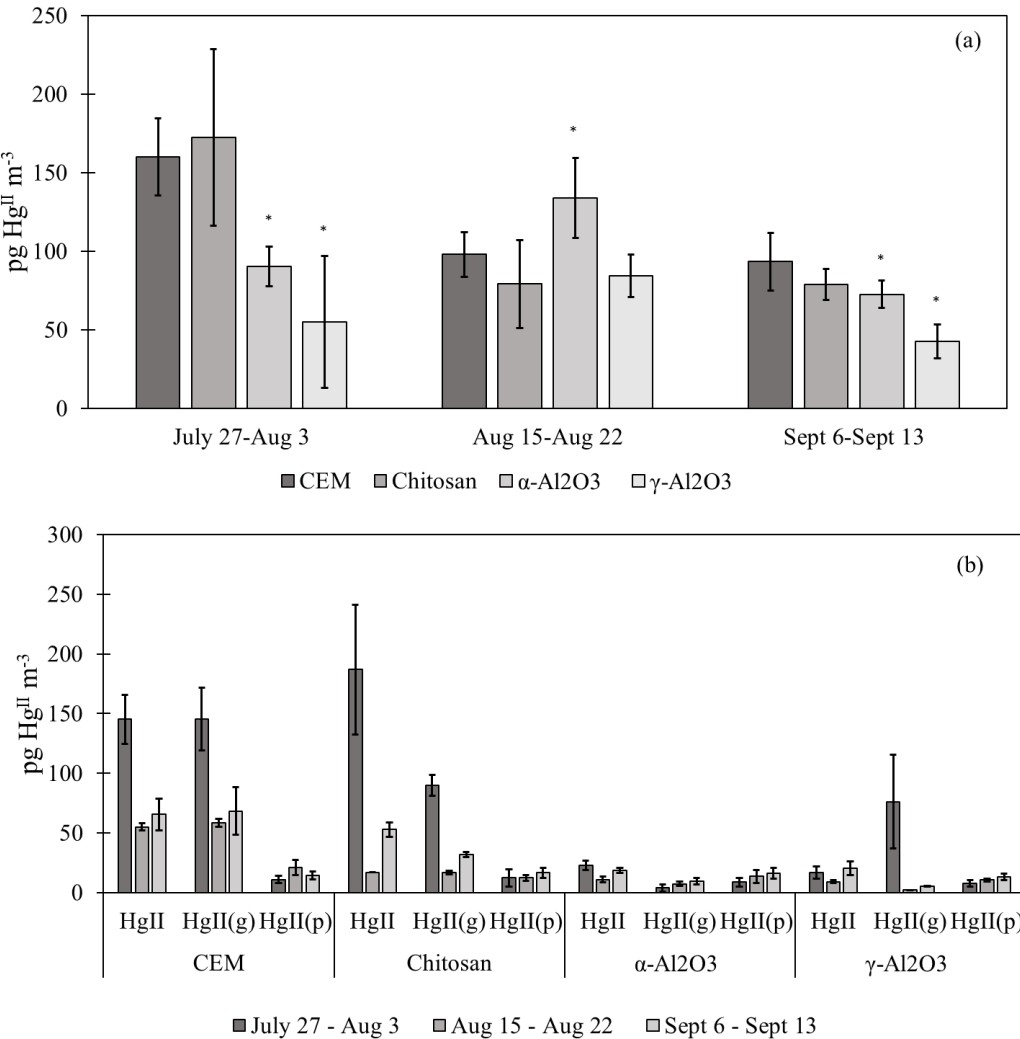

**Fig. H1:** (a) Total Hg$^{II}$ recovery from entire trap assembly (PTFE membrane (if present) + candidate material or first-in-line CEM + breakthrough CEM). Data combine traps with and without upstream PTFE membranes for each material type (n = 6). Error bars represent one standard deviation from the mean. Asterisks (*) indicate a statistically different recovery of Hg$^{II}$ on candidate materials compared to CEM (ANOVA, α ≤ 0.05). (b) Total Hg$^{II}$ recovered from candidate materials or first-in-line CEM without upstream PTFE membranes (Hg$^{II}$; n = 3) shown adjacent to downstream candidate materials or CEM in traps with upstream PTFE (Hg$^{II}_{(g)}$; n = 3) and Hg$^{II}$ recovered from PTFE membranes (Hg$^{II}_{(p)}$). In theory, Hg$^{II}_{(g)}$ + Hg$^{II}_{(p)}$ = Hg$^{II}$.

**Table I1:** Calibrator permeation rates as measured by CEM

| Date | Average (pg s$^{-1}$ ± 1σ) | Number of replicates |
|---|---|---|
| | | |

| | | |
|---|---|---|
| 4/15/2023 | $1.71 \pm 0.35$ | 3 |
| 4/21/2023 | $1.79 \pm 0.08$ | 3 |
| 7/21/2023 | $1.78 \pm 0.06$ | 3 |
| 11/4/2023 | $0.27 \pm 0.09*$ | 3 |
| 11/18/2023 | $0.43 \pm 0.08*$ | 3 |
| 1/18/2024 | $1.82 \pm 0.14$ | 3 |
| 1/25/2024 | $1.83 \pm 0.07$ | 3 |
| 1/29/2024 | $1.71 \pm 0.05$ | 3 |
| 2/15/2024 | $1.76 \pm 0.19$ | 3 |

* $Hg^{II}_{(g)}$ source was not heated during loading.

**Table J1:** pg m$^{-3}$ recoveries of $Hg^{II}$ from traps following field deployment. Columns labeled $Hg^{II}$ and $Hg^{II}$ breakthrough belong to traps deployed without PTFE membranes. Recoveries (pg m$^{-3}$) of $Hg^{II}$ from PTFE are indicated as $Hg^{II}_{(g)}$, recoveries from CEM or candidate material downstream of PTFE are indicated by $Hg^{II}_{(g)}$, and breakthrough from traps with PTFE membranes are indicated by $Hg^{II}_{(g + p)}$ breakthrough.

| Material | $Hg^{II}$ | $Hg^{II}$ breakthrough | $Hg^{II}_{(p)}$ | $Hg^{II}_{(g)}$ | $Hg^{II}_{(p + g)}$ breakthrough |
|---|---|---|---|---|---|
| July 27 - Aug 3 | | | | | |
| CEM | $145 \pm 20$ | $0 \pm 0$ | $11 \pm 3$ | $145 \pm 27$ | $0 \pm 0$ |
| Chitosan | $187 \pm 54$ | $28 \pm 15$ | $12 \pm 7$ | $90 \pm 9$ | $27 \pm 10$ |
| α-Al$_2$O$_3$ | $23 \pm 4$ | $62 \pm 12$ | $9 \pm 4$ | $4 \pm 3$ | $83 \pm 15$ |
| γ -Al$_2$O$_3$ | $17 \pm 5$ | $9 \pm 16$ | $8 \pm 3$ | $76 \pm 39$ | $0 \pm 30$ |
| Aug 15 - Aug 22 | | | | | |
| CEM | $55 \pm 3$ | $0 \pm 0$ | $21 \pm 6$ | $58 \pm 3$ | $0 \pm 0$ |
| Chitosan | $17 \pm 0$ | $42 \pm 13$ | $12 \pm 2$ | $17 \pm 2$ | $70 \pm 27$ |
| α-Al$_2$O$_3$ | $11 \pm 2$ | $115 \pm 14$ | $14 \pm 5$ | $7 \pm 2$ | $121 \pm 27$ |
| γ -Al$_2$O$_3$ | $9 \pm 1$ | $71 \pm 17$ | $11 \pm 1$ | $2 \pm 0$ | $76 \pm 12$ |
| Sept 6 - Sept 13 | | | | | |
| CEM | $65 \pm 13$ | $0 \pm 0$ | $14 \pm 3$ | $68 \pm 20$ | $0 \pm 0$ |
| Chitosan | $53 \pm 6$ | $33 \pm 5$ | $17 \pm 4$ | $32 \pm 2$ | $24 \pm 1$ |
| α-Al$_2$O$_3$ | $19 \pm 2$ | $49 \pm 5$ | $16 \pm 4$ | $10 \pm 3$ | $52 \pm 3$ |
| γ -Al$_2$O$_3$ | $20 \pm 6$ | $27 \pm 8$ | $13 \pm 3$ | $5 \pm 0$ | $20 \pm 4$ |

**Data/code availability**

All data are included in the article and appendix.

**Author contributions**

LL suggested materials, designed and executed the experiments, performed data analysis and interpretation, and prepared the manuscript. SMDC suggested materials, supervised the experiments and data analysis, and edited the manuscript. SNL built and consulted on the use of the $HgBr_2$ calibrator, consulted on other experiments, and edited the manuscript. MSG conceived the project and acquired funding, suggested materials, supervised experiments and edited the manuscript.

**Competing interests**

None of the authors have competing interests.

**Acknowledgements**

This work was funded by the National Science Foundation, Division of Atmospheric and Geospace Sciences, under grants 2043042, 2044537, and 1951513. The authors would like to thank Dr. Igor Slowing for suggesting poly(1,4-phenylene) sulfide in the early stages of the project and Dr. Jan Gačnik for suggesting we invert RMAS shields to deploy powder materials. Thanks also to undergraduate research assistants Mitch Aiken, Nicole Choma, Chris Ford, Ryan Murphy, and Morgan Yeager for help with maintaining the trace-clean glassware and equipment used in this
work. Thanks to the two anonymous reviewers who provided helpful feedback.

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
