# Peer review of "Alternate materials for the capture and quantification of gaseous oxidized mercury in the atmosphere"

_Atmospheric Measurement Techniques, 2024_

## Author Response (AR1)

We thank Referee #1 for careful and thorough consideration of this work, as well as a quick response. Referee #1 provided four areas for improvement (*italicized*), which are addressed individually below. Any changes to the manuscript or appendix are indicated by referenced line and page numbers (in **bold**) which correspond to the line and page numbers of the original submission, rather than the revised manuscript.

*Why were recoveries only based on comparisons to CEM sorption? It would seem relatively easy to include a second set of calculations based on the calibrated HgBr2 salt permeation rates to present a more detailed and comprehensive assessment of the methods. Any such calculations have been dismissed because (1) of differing distance between filter packs and permeation system, and (2) the permeation rate "dropped" during some of the experiments. I don't think those are valid reasons to dismiss those data from analysis (this links for discussion on using CEMs as "real" HgII(g) concentration data below). At minimum, the calculations should be made for the experiments that were not impacted by the short-term permeation malfunction error. I also do not understand why the data on the calculated permeation rates are not included in the appendix/supporting information (see line 166: "data not shown").*

The calibrator used in this study maintains a constant output of $Hg^{II}_{(g)}$ from a permeation tube source that can be validated either gravimetrically (Lyman et al., 2020) or by measurement with CEM (Lyman et al., 2016; Gačnik et al., 2024 ). The gravimetric method requires a highly sensitive balance capable of measuring micrograms of mass loss and at this time our laboratory does not have this specialized equipment. The gravimetric characterization method has recently been shown to measure equivalent permeation rates as a CEM-based dual channel system characterization method, and a gas chromatography mass spectrometry characterization method (Elgiar et al., 2024). Thus, we rely on the CEM method to calculate the permeation rate of the calibrator. Permeation rates are calculated by exposing CEM to calibrator output, then digesting the CEM by EPA Method 1631. The recovered mass of Hg is then divided by the exposure time to provide a pg s$^{-1}$ permeation rate (blank corrected). Since the permeation rate is a CEM-based measurement made by the same method that ng recovery from CEM and alternative materials are measured, comparisons between Hg recovered on alternative materials to recovery on CEM are essentially the same as comparisons to the calculated permeation rate, since both CEM and alternative materials are exposed to calibrator output for the same amount of time.

Previous work loading CEM with $Hg^{II}_{(g)}$ in the laboratory demonstrated that $Hg^{II}_{(g)}$ is quantitatively retained on CEM. For instance, Miller et al., (2019) loaded $HgBr_2$ onto CEM while simultaneously measuring the permeation source output using a dual channel system. They found a 123% recovery of permeated $Hg^{II}_{(g)}$ compared to the same permeation source output measured by a Tekran 2573. Another study, using the same dual channel system design to test CEM, observed negligible breakthrough and linear sorption of $HgBr_2$ with masses of up to 3 ng, which is three times higher than the masses loaded in this study (Dunham-Cheatham et al., 2020). These authors also observed quantitative retention of ambient $Hg^{II}$ for 190 days. All materials in this study were analyzed in fewer than seven days from sample collection. Elgiar et al. (2024) have also demonstrated that CEM-based dual channel measurements of $Hg^{II}$ output from an SI-traceable, gravimetrically-characterized, permeation source were accurate in a field setting. This further suggests that CEM accurately quantify $Hg^{II}$.

During replication of alternate material $Hg^{II}$ exposure experiments, the heated chamber in the calibrator cooled, causing a drop in the permeation rate (as measured by CEM in each experiment). As a result, a different total mass of Hg was loaded across replicate experiments. To make data comparable across replicates, each experiment was normalized to the CEM control for that experiment.

**Several changes were made to the text and appendix to clarify these points.**

          **Line 154, page 5, was clarified as:** "Sorption of $Hg^{II}_{(g)}$ to candidate materials was performed with the same procedure, using a custom-built $HgBr_2$ calibrator (Allen et al., 2024; Gačnik et al., 2024) that releases a constant stream of $Hg^{II}_{(g)}$ from a salt-based permeation source. The permeation rate is tightly controlled by maintaining constant temperature, pressure, and He flow over the permeation source. The permeation rate of the calibrator can be determined either gravimetrically or by measurement with CEM (Lyman et al., 2016; Gačnik et al., 2024). These methods have been demonstrated to be equivalent by Elgiar et al. (2024). The CEM method was used to calculate the permeation rate of the calibrator in this study. Briefly, permeation rates are calculated by exposing CEM to calibrator output, then digesting the CEM by EPA Method 1631. The recovered mass of Hg is then divided by the exposure time to provide a pg s$^{-1}$ permeation rate (blank corrected)."

          **Line 163, page 5, was amended as:** "Returning the heated chamber to 50 °C restored the measured permeation rate to 1.77 ± 0.06 (data available in Appendix Table I1)."

          **Line 167, page 5, has been clarified as:** "Due to the change in permeation rate across replicate experiments, results are reported as a % of $Hg^{II}_{(g)}$ recovered from CEM, rather than a % of the expected recovery based on the perm rate that was calculated as the mass of $Hg^{II}$ recovered from CEM divided by exposure time (pg s$^{-1}$)."

          **Measured permeation rates were also added to the appendix on Line 24, page 19:**

Table I1: Calibrator permeation rates as measured by CEM

| Date | Average (pg s$^{-1}$ ± 1σ) | Number of replicates |
|---|---|---|
| 4/15/2023 | 1.71 ± 0.35 | 3 |
| 4/21/2023 | 1.79 ± 0.08 | 3 |
| 7/21/2023 | 1.78 ± 0.06 | 3 |
| 11/4/2023 | 0.27 ± 0.09* | 3 |
| 11/18/2023 | 0.43 ± 0.08* | 3 |
| 1/18/2024 | 1.82 ± 0.14 | 3 |
| 1/25/2024 | 1.83 ± 0.07 | 3 |
| 1/29/2024 | 1.71 ± 0.05 | 3 |
| 2/15/2024 | 1.76 ± 0.19 | 3 |

* $Hg^{II}_{(g)}$ source was not heated during loading.

**A discussion of quantitative Hg$^{II}$ sorption by CEM was also included in the text, addressed in comments below.**

*Following on from this, I also question whether using BrCl2 salt is really a valid way to assess HgII(g) sorption? HgBr2 is one of potentially 100s of different HgII(g) species. Do they all behave the same? I understand that these measurements are difficult to calibrate and evaluate due to the dearth of effective measurement systems and calibration standards, but much more discussion is needed on this. Furthermore, the issues with variable permeation rates due to system issues, distance from permeation unit (and concerns of representativeness) do bring up further questions relating to the effectiveness of using this system as the sole means of calibrating GOM measurement systems.*

The reviewer is correct, there is a great deal of uncertainty surrounding exactly what Hg$^{II}$ compounds are present in the atmosphere, because no methods currently exist that can observe the chemical identity of these compounds. This makes it difficult to identify what the most valid Hg compound to test might be. Some of the most recently considered species (Shah et al., 2021; Hewa et al., 2023) are not commercially available, although halogen radicals are generally accepted atmospheric Hg$^0$ oxidants based on both modeling (Holms et al., 2010; Horowitz et al., 2017; Song et al., 2024) and experimental observations of atmospheric Hg depletion events in the Arctic (Steffen et al., 2008), as well as formation of Hg$^{II}$ at the marine boundary layer (Laurier et al., 2003). The GEOS-Chem (Gustin et al., 2023) and Shah et al. (2021) models indicate HgCl$_2$ comprises a substantial portion of Hg$^{II}$ in the atmosphere, although this remains to be experimentally verified. This suggests that HgBr$_2$ or HgCl$_2$ are atmospherically relevant Hg$^{II}$ compounds to test and are widely used for Hg$^{II}_{(g)}$ measurement method development (McClure et al., 2014; Lyman et al., 2020, Gačnik et al., 2022; Dunham-Cheatham et al., 2023; Jones et al., 2016).

The authors agree that gaseous Hg compounds likely do not all behave the same. Additional testing with a range of representative compounds should be performed for any promising new sorptive surface before it can be deployed in the field with confidence, as stated on Lines 305 and 306. Laboratory and field tests demonstrated that these materials did not collect ambient Hg$^{II}$ quantitatively and thus, testing with additional compounds was not necessary. This was demonstrated by the observed decrease in Hg$^{II}$ recovery by all tested alternative materials compared to CEM in the field, reported in Figure 2a.

Two points are made by Referee #1 regarding the reliability of this calibration system: 1. A drop in permeation rate was reported during experimentation; and 2. CEM distance from the permeation unit appears to affect the calculated permeation rate. Hopefully, the inclusion of measured permeation rates in the appendix will resolve the first concern, as these data demonstrate the drop in permeation was indicative of user error (the chamber temp can be controlled through the user interface, but was not reset during the experiments in question), rather than indicative of the calibrator's actual performance. To the second concern, the correlation between distance from the calibrator and CEM recovery is interesting. Previous work has shown that less than 5% of atmospheric Hg$^{II}_{(g)}$ is lost to the PTFE filterpacks (Allen et al., 2024) during field sampling. However, as Referee #1 points out, it is possible that the Hg$^{II}_{(g)}$ permeated from the calibrator may behave differently from atmospheric Hg$^{II}_{(g)}$. The use of this

particular calibrator here, and in Allen et al. (2024), are the first instances of an independent laboratory employing the calibrator since its development at Utah State University, and is part of a broader effort to validate its performance as a calibration system, before it can become more widely available to the research community. Validation of the permeation rate by gravimetric methods, as well as an investigation of $HgBr_2$ loss to PTFE filterpacks, would clarify the concern and is planned for future work; but is outside the scope of this paper. The purpose of this work was to compare $Hg^{II}$ recovery from uncharacterized materials to the widely used CEM when exposed to the same quantity of $Hg^{II}$ from the same source, rather than to calibrate any measurement system. The term "calibrator" maybe causing some confusion here and will be clarified in the text. The instrument was developed to calibrate $Hg^{II}_{(g)}$ measurements, but was used here to load materials with a single, measurable quantity of $Hg^{II}$. The permeation source inside the calibrator is held in a chamber at a constant temperature and pressure. Helium flows over the perm tube through the chamber to maintain a constant atmospheric composition, even while not in use. The tightly controlled permeation source in the calibrator makes it ideal for this comparison.

**The introduction was rewritten to include a discussion of our current understanding of elemental Hg oxidation in the atmosphere and the use of mercuric halogens as model $Hg^{II}_{(g)}$ compounds. The likely difference in behavior between different $Hg^{II}$ compounds was also further acknowledged.**

**Beginning on Line 32, page 1, the introduction now reads:** "A complete understanding of Hg behavior in the atmosphere is necessary to describe the fate of anthropogenic Hg pollution, assess health risks to humans and wildlife, and evaluate the effectiveness of the Minamata Convention. The mechanisms that govern the oxidation and reduction of Hg in the atmosphere are not well understood (Shah et al., 2021), and model results are uncertain, because they do not consider other forms of $Hg^{II}$ present (Gustin et al., 2023). For example, Shah et al. (2021) assumed all $Hg^{II}$ compounds devolatilized from aerosols are $HgCl_2$. This is an assumption that has not been validated.

Currently, Br- and Cl- radicals are considered to participate in elemental Hg ($Hg^0$) oxidation. This is based on both theoretical work (Holms et al., 2010; Horowitz et al., 2017; Song et al., 2024) and experimental observations of atmospheric Hg depletion events in the Arctic (Steffen et al., 2008), as well as observations of $Hg^{II}$ formation at the marine boundary layer (Laurier et al., 2003). The identity of $Hg^{II}_{(g)}$ compounds in the atmosphere is currently unknown, but mass spectrometry (MS) methods capable of observing atmospheric $Hg^{II}_{(g)}$ speciation are in development (cf., Jones et al., 2016; Khalizov et al., 2020; Mao and Khalizov, 2021). These methods have been developed using $HgBr_2$ and $HgCl_2$ as model atmospheric $Hg^{II}_{(g)}$ compounds, given the role of halogen radicals in atmospheric $Hg^0$ oxidation. However, due to differences in $Hg^{II}_{(g)}$ behavior, the use of a broad range of representative compounds is desirable in both MS method development and validation of preconcentration surfaces."

**Clarification about the use of the $Hg^{II}$ "calibrator" was also added to Line 93, page 3:** "A custom-built $Hg^{II}$ permeation calibrator was used to load candidate materials with a known quantity of $Hg^{II}$, for comparison to CEM. $Hg^{II}$ capture by these materials was also compared under field conditions.", **and Line 156, page 5:** "Although this calibrator can be used to calibrate $Hg^{II}_{(g)}$ measurements in other systems, it was used here to compare sorptive properties of candidate materials to CEM, by delivering consistent, measurable, quantities of $Hg^{II}_{(g)}$"

**The discussion of differing permeation rates was expanded on Line 161, page 5:** "The difference in observed permeation rate between these two studies is of significance for the use of this system for calibrating $Hg^{II}$ measurements and should be studied further before it is broadly employed by the research community. A possible explanation for the difference may be the positioning of the calibrator tip at a distance of 2 cm from the CEM during loading (Gačnik et al., 2024) versus at the filter pack inlet (5.5 cm in this work), as $HgBr_2$ is more likely to come in contact with the filterpack when loaded at the inlet. Work by Allen et al. (2024) suggest less than 5% of atmospheric $Hg^{II}$ is sorbed to the PTFE filterpacks after field deployment."

*Another oversight I see is the limited information on the blank (unexposed) Hg concentrations of these surfaces, and in particular the Hg concentrations on the unexposed CEMs. Previous studies have shown these blanks can have elevated and variable Hg concentrations before exposure. Since the whole study is premised on the CEMs providing the "real" HgII(g) concentration data, should there not be a very clear assessment of the CEM blank levels? Is there no pre-cleaning (acid bath) method for CEMs to improve blank levels to ensure what is measured comes from the deployments. If CEMs present "real" HgII(g) concentration data (something that I do not believe has been comprehensively confirmed in the literature or by the wider atmospheric Hg science community), then surely very complete QAQC data for this method must be detailed.*

Blank CEM were analyzed with $HgBr_2$-exposed CEM in every experiment and used to blank-correct the reported CEM recoveries. Historically, this laboratory has had very low total Hg recoveries on blank CEMs (mean of 53.4 ± 60.7 pg Hg per membrane; ± 1 standard deviation; n = 133, reported by Dunham-Cheatham et al., 2023; available in the appendix). For this project in particular, the mean recovery on unexposed 47 mm diameter CEM was 30 ± 10 pg (± 1 standard deviation) across 28 membranes. This level of contamination is at the limit of quantification of our most sensitive analysis (25 pg). Similarly, blanks were analyzed for each of the alternative materials tested. Mean recovery was 10 ± 20, 10 ± 10, and 20 ± 10 pg per material mass (30 mg) of $\alpha$-$Al_2O_3$, $\gamma$-$Al_2O_3$ and chitosan, respectively. The accuracy of $Hg^{II}$ measurement by CEM is addressed under the first comment, as well as in the following comment.

**These data have been added to the paper on Line 175, page 5:** "Blanks for each tested material were collected and analyzed with samples during each experiment, and were used to correct the analyzed value of samples. These blanks were exposed to laboratory air only (no $HgBr_2$) during laboratory $HgBr_2$ exposure tests, or were not exposed to air during field campaigns. Mean recovery on CEM blanks (exposed to laboratory air or not) was 0.03 ± 0.01 ng per membrane (± 1 standard deviation) across 28 replicates. Mean recovery of $\alpha$-$Al_2O_3$, $\gamma$-$Al_2O_3$ and chitosan was 0.01 ± 0.02, 0.01 ± 0.01, and 0.02 ± 0.01 ng per target mass (30 mg), respectively (n = 27 for each material)."

*In some ways this also leads to the question: if CEMs are effective HgII(g) measurements sorbents, then why do alternative capture methods need to be tested. I guess the answer to this question lies in the discussion on lines 45-79 (concerns of reduced CEM capture efficiency and surface reaction), which loops back to concerns of CEMs being considered the "real" data.*

This work seeks to test new materials that can preconcentrate $Hg^{II}$ for thermal desorption into mass spectrometry (MS) systems to determine the chemistry of $Hg^{II}$ compounds (Lines 91-94). CEM are used to quantify $Hg^{II}$ concentrations and are analyzed by EPA Method 1631. MS methods are currently under development, but are expected to utilize thermal desorption for field sample introduction into the instrument. Although CEM provide a quantitative $Hg^{II}$ measurement, they are not appropriate for thermal desorption, as they are known to generate compounds that interfere with Hg quantification when heated (Gustin et al., 2019) and likely facilitate exchange reactions with Hg compounds on the CEM surface (Mao and Khalizov, 2021). Thus, alternative materials are needed for this application, and CEM serve as a useful benchmark for comparing sorption performance. As mentioned under the first

comment, CEM are excellent sorptive surfaces for standard salts under laboratory conditions (Miller et al., 2019; Dunham-Cheatham et al., 2020); however, field conditions can be significantly different and highly variable. Important differences include longer sampling times and the presence of particulates and reactive gases, like ozone and humidity. Variable environmental conditions may explain, in part, the observed decrease in CEM performance compared to dual channel systems, discussed on Lines 48-56. Some evidence for increased capture of $Hg^{II}$ by CEM has already been presented in the literature under high humidity conditions in the laboratory (Huang and Gustin, 2015) and field (Bu et al., 2018). However, CEM have performed well compared to other available measurement methods in the field (Gustin et al., 2019; Bu et al., 2018), and remain a useful comparative tool for new materials. The performance of CEM in the field is an active area of research, and there have been reports which show quantitative measurement of $Hg^{II}$ in the field by CEM-based dual channel systems when compared to other methods (Elgiar et al., 2024; Lyman et al., 2020).

**Additional detail was added to Line 47, page 2:** "Although CEM outperform KCl denuders for quantitative $Hg^{II}_{(g)}$ capture (Huang et al., 2013), and are quantitative $Hg^{II}_{(g)}$ sorbants under laboratory conditions (Miller et al., 2019, Dunham-Cheatham 2020), recent work suggests CEM may not be fully quantitative under field conditions. For instance, Dunham-Cheatham et al. (2023)…" **and Line 56, page 2:** "CEM are not appropriate for $Hg^{II}$ sample introduction into MS systems, for when heated they generate compounds that interfere with Hg quantification by cold vapor atomic absorption spectroscopy (Gustin et al., 2019), and research done using high $Hg^{II}$ concentrations has suggested that exchange reactions can occur with Hg compounds on the CEM surface (Mao and Khalizov, 2021)."

**Line 305, page 11, was clarified:** "Promising materials should be tested for: sorption of $Hg^{II}_{(g)}$ and $Hg^0$ capture efficiency for a broad range of representative Hg compounds (Dunham-Cheatham et al., 2020); the potential for chemical transformation on the material surface; potential reactions between the Hg sample and other atmospheric constituents, including interferences with humidity (Huang and Gustin, 2015) and ozone (McClure et al., 2014); and for performance under both laboratory and field conditions."

*In lines 275-278, the authors provide discussion on the possible capture of a small fraction of HgII(g) species on PTFE filters (used to remove particulates and HgII(p)). They then use this as justification to add ALL HgII(p) captured on the PTFE filters to the HgII(g) measurements of the sorbents tested in these experiments. I do not believe this is valid and appears to be a means to increase sorbent recoveries. I do not question that some HgII(g) could sorb to particles attached to PFTE filters - this is a legitimate concern, but I do not agree that all HgII(p) from the PTFE should be added to the HgII(g) sorbent data. Indeed, delving into the Allen et al. (2024) paper, they state PTFE captures only 4% of HgII(g) (described as GOM in that study) on new PTFE filters and ~50% on filters previously deployed for 1 week. Based on the methods in this manuscript, it appears the same brand and new PTFE filters were utilized. Therefore, there is no justification for assuming all the HgII(p) captured on PTFE is HgII(g) that would have been captured by the HgII(g) sorbents tested. All the HgII(p) data from the PTFE filters should be included in the appendix/supporting information if not already.*

$Hg^{II}_{(p)}$ and $Hg^{II}_{(g)}$ are operationally defined terms for particulate-bound and gaseous oxidized mercury, respectively. The sum of $Hg^{II}_{(p)}$ and $Hg^{II}_{(g)}$ measurements have been found to be similar to $Hg^{II}$

measurements in recent field studies (Gustin et al., 2019; Gustin et al., 2023). The data presented in Allen et al. (2024) demonstrate that these definitions (determined by the pore size of the membrane) may not be as physically meaningful as previously thought, as a significant amount of $Hg^{II}_{(g)}$ is captured by particulates on PTFE membranes, highlighted by Referee #1. This study employed the same membranes, in the same configuration, so the distinction between $Hg^{II}_{(p)}$ and $Hg^{II}_{(g)}$ was not considered meaningful since an unquantifiable amount of the $Hg^{II}_{(p)}$ is likely $Hg^{II}_{(g)}$. Thus, recovery from PTFE filters was added to the $Hg^{II}_{(g)}$ recovery from downstream CEM or candidate materials and the sum was considered a total oxidized Hg measurement ($Hg^{II}$). This removes any assumption about the gaseous vs particulate-bound form of the recovered Hg, and is more technically correct. Total Hg measurements for candidate materials were compared to total Hg measurements ($Hg^{II}_{(g)}$ + $Hg^{II}_{(p)}$) from CEM deployed concurrently. Since the PTFE data ("$Hg^{II}_{(p)}$") was added to both the downstream candidate material traps and downstream CEM filterpacks, recoveries are increased for both candidate materials and CEM. This does not change the evaluation of candidate material sorption compared to CEM and results are discussed only as $Hg^{II}$, rather than $Hg^{II}_{(g)}$, removing any assumption about the form of $Hg^{II}$ recovered. A typo on line 283 where $Hg^{II}$ was written as $Hg^{II}_{(g)}$ was identified and corrected. Recoveries from PTFE filters ("$Hg^{II}_{(p)}$") and from downstream CEM and candidate materials ("$Hg^{II}_{(g)}$") are already presented separately in Figure H1 of the appendix. At Referee #2's request, the numeric values of recoveries presented in Figure H1 have also been presented as a table in the appendix (Table J1), with the addition of breakthrough recoveries. No assumption about $Hg^{II}_{(p)}$ or $Hg^{II}_{(g)}$ capture was made about the candidate materials. Breakthrough from each filterpack or trap was measured using CEM to determine if candidate materials were indeed capturing $Hg^{II}$. These results are presented in Figure 2b.

**This was clarified in the text on the following lines:**

**Line 271, page 9:** "Field measurements included a downstream CEM that captured $Hg^{II}$ not sorbed by candidate materials (i.e. "breakthrough"), if present."

**Line 283, page 10 was corrected:** "…indicating that chitosan, α-$Al_2O_3$, and γ-$Al_2O_3$ did not quantitatively measure $Hg^{II}$ under field conditions."

**The measurements displayed in Fig. 2b are now available as a table in the appendix.**

Table J1: pg $m^{-3}$ recoveries of $Hg^{II}$ from traps following field deployment. Columns labeled $Hg^{II}$ and $Hg^{II}$ breakthrough belong to traps deployed without PTFE membranes. Recoveries (pg $m^{-3}$) of $Hg^{II}$ from PTFE are indicated as $Hg^{II}_{(g)}$, recoveries from CEM or candidate material downstream of PTFE are indicated by $Hg^{II}_{(g)}$, and breakthrough from traps with PTFE membranes are indicated by $Hg^{II}_{(g+p)}$ breakthrough.

| Material | $Hg^{II}$ | $Hg^{II}$ breakthrough | $Hg^{II}_{(p)}$ | $Hg^{II}_{(g)}$ | $Hg^{II}_{(p+g)}$ breakthrough |
|---|---|---|---|---|---|
| July 27 - Aug 3 | | | | | |
| CEM | 145 ± 20 | 0 ± 0 | 11 ± 3 | 145 ± 27 | 0 ± 0 |
| Chitosan | 187 ± 54 | 28 ± 15 | 12 ± 7 | 90 ± 9 | 27 ± 10 |
| α-$Al_2O_3$ | 23 ± 4 | 62 ± 12 | 9 ± 4 | 4 ± 3 | 83 ± 15 |

| | | | | | |
|---|---|---|---|---|---|
| γ -Al$_2$O$_3$ | 17 ± 5 | 9 ± 16 | 8 ± 3 | 76 ± 39 | 0 ± 30 |
| Aug 15 - Aug 22 | | | | | |
| CEM | 55 ± 3 | 0 ± 0 | 21 ± 6 | 58 ± 3 | 0 ± 0 |
| Chitosan | 17 ± 0 | 42 ± 13 | 12 ± 2 | 17 ± 2 | 70 ± 27 |
| α-Al$_2$O$_3$ | 11 ± 2 | 115 ± 14 | 14 ± 5 | 7 ± 2 | 121 ± 27 |
| γ -Al$_2$O$_3$ | 9 ± 1 | 71 ± 17 | 11 ± 1 | 2 ± 0 | 76 ± 12 |
| Sept 6 - Sept 13 | | | | | |
| CEM | 65 ± 13 | 0 ± 0 | 14 ± 3 | 68 ± 20 | 0 ± 0 |
| Chitosan | 53 ± 6 | 33 ± 5 | 17 ± 4 | 32 ± 2 | 24 ± 1 |
| α-Al$_2$O$_3$ | 19 ± 2 | 49 ± 5 | 16 ± 4 | 10 ± 3 | 52 ± 3 |
| γ -Al$_2$O$_3$ | 20 ± 6 | 27 ± 8 | 13 ± 3 | 5 ± 0 | 20 ± 4 |

We thank Referee #2 for their time and a thoughtful review. Referee #2 provided five comments (*italicized*), which are addressed individually below, and were similar in nature to comments made by Referee #1. Below are responses to Referee #2 that include significant portions of responses made to Referee #1, which were not available to Referee #2 at the time. Responses have been elaborated upon where appropriate. Any changes to the manuscript or appendix are indicated by referenced line and page numbers (in **bold**) which correspond to the line and page numbers of the original submission, rather than the revised manuscript.

*The authors assessed the three candidate materials by comparing their Hg$^{II}$ recovery rates to that of CEM, but the latter was not shown and thus, it seemed to be assumed that the recovery rate of CEM would be 100%. If it were true, then why bother to find more materials? Why didn't the authors provide "real" Hg$^{II}$ recovery rates for the three candidate materials?*

The recovery rate of HgBr$_2$ from CEM was assumed to be 100% in this study. This work seeks to test new materials that can preconcentrate Hg$^{II}$ for thermal desorption into mass spectrometry (MS) systems to determine the chemistry of Hg$^{II}$ compounds (Lines 91-94). CEM are used to quantify Hg$^{II}$ concentrations and are analyzed by EPA Method 1631. MS methods are currently under development, but are expected to utilize thermal desorption for field sample introduction into the instrument. Although CEM provide a quantitative Hg$^{II}$ measurement, they are not appropriate for thermal desorption, as they are known to generate compounds that interfere with Hg quantification when heated (Gustin et al., 2019) and likely facilitate exchange reactions with Hg compounds on the CEM surface (Mao and Khalizov, 2021). Thus, alternative materials are needed for this application, and CEM serve as a useful benchmark for comparing sorption performance.

Previous work loading CEM with Hg$^{II}_{(g)}$ in the laboratory demonstrated that Hg$^{II}_{(g)}$ is quantitatively retained on CEM. For instance, Miller et al., (2019) loaded HgBr$_2$ onto CEM while simultaneously

measuring the permeation source output using a dual channel system. They found a 123% recovery of permeated $Hg^{II}_{(g)}$ compared to the same permeation source output measured by a Tekran 2573. Another study, using the same dual channel system design to test CEM, observed negligible breakthrough and linear sorption of $HgBr_2$ with masses of up to 3 ng, which is three times higher than the masses loaded in this study (Dunham-Cheatham et al., 2020). These authors also observed quantitative retention of ambient $Hg^{II}$ for 190 days. All materials in this study were analyzed in fewer than seven days from sample collection. Elgiar et al. (2024) have also demonstrated that CEM-based dual channel measurements of $Hg^{II}$ output from an SI-traceable, gravimetrically characterized, permeation source were accurate in a field setting. This further suggests that CEM accurately quantify $Hg^{II}$.

The $Hg^{II}$ recovered from candidate materials is presented in this study as a percentage of the mass $Hg^{II}$ recovered from CEM. The calibrator used in this study maintains a constant output of $Hg^{II}_{(g)}$ from a permeation tube source that can be validated either gravimetrically (Lyman et al., 2020) or by measurement with CEM (Lyman et al., 2016; Gačnik et al., 2024 ). The gravimetric method requires a highly sensitive balance capable of measuring micrograms of mass loss and at this time our laboratory does not have this specialized equipment. The Elgiar et al. (2024) paper demonstrated that these methods are equivalent, and thus, we rely on the CEM method to calculate the permeation rate of the calibrator. Permeation rates are calculated by exposing CEM to calibrator output, then digesting the CEM by EPA 1631. The recovered mass of Hg is then divided by the exposure time to provide a pg s$^{-1}$ permeation rate (blank corrected). Since the permeation rate is a CEM-based measurement made by the same method that ng recovery from CEM and alternative materials are measured, comparisons between Hg recovered on alternative materials to recovery on CEM are essentially the same as comparisons to the calculated permeation rate, since both CEM and alternative materials are exposed to calibrator output for the same amount of time.

**Several changes were made to the text and appendix to clarify these points.**

      **Additional detail was added to Line 47, page 2: "**Although CEM outperform KCl denuders for quantitative $Hg^{II}_{(g)}$ capture (Huang et al., 2013), and are quantitative $Hg^{II}_{(g)}$ sorbents under laboratory conditions (Miller et al., 2019, Dunham-Cheatham 2020), recent work suggests CEM may not be fully quantitative under field conditions. For instance, Dunham-Cheatham et al. (2023)**" and Line 56, page 2:** "CEM are not appropriate for $Hg^{II}$ sample introduction into MS systems, for when heated they generate compounds that interfere with Hg quantification by cold vapor atomic absorption spectroscopy (Gustin et al., 2019), and research done using high $Hg^{II}$ concentrations has suggested that exchange reactions can occur with Hg compounds on the CEM surface (Mao and Khalizov, 2021)."

      **Line 154, page 5, was clarified as:** "Sorption of $Hg^{II}_{(g)}$ to candidate materials was performed with the same procedure, using a custom-built $HgBr_2$ calibrator (Allen et al., 2024; Gačnik et al., 2024) that releases a constant stream of $Hg^{II}_{(g)}$ from a salt-based permeation source. The permeation rate is tightly controlled by maintaining constant temperature, pressure, and He flow over the permeation source. The permeation rate of the calibrator can be determined either gravimetrically or by measurement with CEM (Lyman et al., 2016; Gačnik et al., 2024). These methods have been demonstrated to be equivalent by Elgiar et al. (2024). The CEM method was used to calculate the permeation rate of the calibrator in this study. Briefly, permeation rates are calculated by exposing CEM to calibrator output, then digesting the CEM by EPA Method 1631. The recovered mass of Hg is then divided by the exposure time to provide a pg s$^{-1}$ permeation rate (blank corrected)."

**Line 166, page 5, has been clarified as:** "Due to the change in permeation rate across replicate experiments, results are reported as a % of $Hg^{II}_{(g)}$ recovered from CEM, rather than a % of the expected recovery based on the perm rate that was calculated as the mass of $Hg^{II}$ recovered from CEM divided by exposure time (pg s$^{-1}$)."

*Why did the permeation rates drop nearly by a factor of 5? Please show all the experimental data to make their point.*

The permeation rate of the $HgBr_2$ calibrator dropped due to a change in the permeation chamber temperature that was not identified until after the experiments had been performed (lines 164-165). When the permeation chamber was returned to 50 °C, the permeation rate also returned to its previous performance. These data are now available in the appendix (Table I1).

**Line 166, page 6, was amended as:** "Returning the heated chamber to 50 °C restored the measured permeation rate to 1.77 ± 0.06 (data available in Appendix Table I1)."

Table I1: Calibrator permeation rates as measured by CEM

| Date | Average (pg s$^{-1}$ ± 1σ) | Number of replicates |
|---|---|---|
| 4/15/2023 | 1.71 ± 0.35 | 3 |
| 4/21/2023 | 1.79 ± 0.08 | 3 |
| 7/21/2023 | 1.78 ± 0.06 | 3 |
| 11/4/2023 | 0.27 ± 0.09* | 3 |
| 11/18/2023 | 0.43 ± 0.08* | 3 |
| 1/18/2024 | 1.82 ± 0.14 | 3 |
| 1/25/2024 | 1.83 ± 0.07 | 3 |
| 1/29/2024 | 1.71 ± 0.05 | 3 |
| 2/15/2024 | 1.76 ± 0.19 | 3 |

*$Hg^{II}_{(g)}$ source was not heated during loading.

*In Fig. 2b, the percentage values were shown, but how in absolute values did the amounts of $Hg^{II}$ captured on breakthrough CEM, candidate material, and PTFE compare to what CEM and*

*breakthrough CEM captured? I am aware the authors showed absolute amounts in Fig. H1, but it was difficult to gauge. Could they please show a table of their measurement data for the figure?*

**The measurements displayed in Fig. 2b are now available as a table in the appendix, with the addition of breakthrough measurements:**

Table J1: pg m$^{-3}$ recoveries of Hg$^{II}$ from traps following field deployment. Columns labeled Hg$^{II}$ and Hg$^{II}$ breakthrough belong to traps deployed without PTFE membranes. Recoveries (pg m$^{-3}$) of Hg$^{II}$ from PTFE are indicated as Hg$^{II}_{(g)}$, recoveries from CEM or candidate material downstream of PTFE are indicated by Hg$^{II}_{(g)}$, and breakthrough from traps with PTFE membranes are indicated by Hg$^{II}_{(g+p)}$ breakthrough.

| Material | Hg$^{II}$ | Hg$^{II}$ breakthrough | Hg$^{II}_{(p)}$ | Hg$^{II}_{(g)}$ | Hg$^{II}_{(p+g)}$ breakthrough |
|---|---|---|---|---|---|
| July 27 - Aug 3 | | | | | |
| CEM | 145 ± 20 | 0 ± 0 | 11 ± 3 | 145 ± 27 | 0 ± 0 |
| Chitosan | 187 ± 54 | 28 ± 15 | 12 ± 7 | 90 ± 9 | 27 ± 10 |
| α-Al$_2$O$_3$ | 23 ± 4 | 62 ± 12 | 9 ± 4 | 4 ± 3 | 83 ± 15 |
| γ -Al$_2$O$_3$ | 17 ± 5 | 9 ± 16 | 8 ± 3 | 76 ± 39 | 0 ± 30 |
| Aug 15 - Aug 22 | | | | | |
| CEM | 55 ± 3 | 0 ± 0 | 21 ± 6 | 58 ± 3 | 0 ± 0 |
| Chitosan | 17 ± 0 | 42 ± 13 | 12 ± 2 | 17 ± 2 | 70 ± 27 |
| α-Al$_2$O$_3$ | 11 ± 2 | 115 ± 14 | 14 ± 5 | 7 ± 2 | 121 ± 27 |
| γ -Al$_2$O$_3$ | 9 ± 1 | 71 ± 17 | 11 ± 1 | 2 ± 0 | 76 ± 12 |
| Sept 6 - Sept 13 | | | | | |
| CEM | 65 ± 13 | 0 ± 0 | 14 ± 3 | 68 ± 20 | 0 ± 0 |
| Chitosan | 53 ± 6 | 33 ± 5 | 17 ± 4 | 32 ± 2 | 24 ± 1 |
| α-Al$_2$O$_3$ | 19 ± 2 | 49 ± 5 | 16 ± 4 | 10 ± 3 | 52 ± 3 |
| γ -Al$_2$O$_3$ | 20 ± 6 | 27 ± 8 | 13 ± 3 | 5 ± 0 | 20 ± 4 |

*I am not convinced that the Hg$^{II}$(p) collected on PTFE should be added to the captured Hg$^{II}$(g) on the candidate materials, unless the amount of Hg$^{II}$(g) adsorbed to particles was demonstrated to be comparable compared to the latter. Citing "the sum of HgII(p) + HgII(g) recovered from PTFE + CEM, respectively, has been well correlated with HgII measurements on CEM in previous work (Gustin et al., 2019)" was not sufficient as the statement did not reveal anything mechanistic.*

Hg$^{II}_{(p)}$ and Hg$^{II}_{(g)}$ are operationally defined terms for particulate-bound and gaseous oxidized mercury, respectively. The sum of Hg$^{II}_{(p)}$ and Hg$^{II}_{(g)}$ measurements have been found to be similar to Hg$^{II}$ measurements in recent field studies (Gustin et al., 2019; Gustin et al., 2023). The data presented in

Allen et al. (2024) demonstrate that these definitions (determined by the pore size of the membrane) may not be as physically meaningful as previously thought, as a significant amount of $Hg^{II}_{(g)}$ is captured by particulates on PTFE membranes. This study employed the same membranes, in the same configuration, so the distinction between $Hg^{II}_{(p)}$ and $Hg^{II}_{(g)}$ was not considered meaningful since an unquantifiable amount of the $Hg^{II}_{(p)}$ is likely $Hg^{II}_{(g)}$. Thus, recovery from PTFE filters was added to the $Hg^{II}_{(g)}$ recovery from downstream CEM or candidate materials and the sum was considered a total oxidized Hg measurement ($Hg^{II}$). This removes any assumption about the gaseous vs particulate-bound form of the recovered Hg, and is more technically correct. Total Hg measurements for candidate materials were compared to total Hg measurements ($Hg^{II}_{(g)}$ + $Hg^{II}_{(p)}$) from CEM deployed concurrently. Since the PTFE data ("$Hg^{II}_{(p)}$") were added to both the downstream candidate material traps and downstream CEM filterpacks, recoveries are increased for both candidate materials and CEM. This does not change the evaluation of candidate material sorption compared to CEM and results are discussed only as $Hg^{II}$, rather than $Hg^{II}_{(g)}$, removing any assumption about the form of $Hg^{II}$ recovered. A typo on line 283 where $Hg^{II}$ was written as $Hg^{II}_{(g)}$ was identified and corrected.

**This was clarified in the text on the following lines:**

**Line 283, page 10 was corrected:** "…indicating that chitosan, $\alpha$-$Al_2O_3$, and $\gamma$-$Al_2O_3$ did not quantitatively measure $Hg^{II}$ under field conditions."

*The four materials measured a wide range of $Hg^{II}$ species in the field, whereas the one, $HgBr_2$, used in the lab for recovery experiments was merely one of numerous $Hg^{II}$ species in the real atmosphere. How would the recovery rates from the field be interpreted in comparison to the lab-based rates?*

This study used $HgBr_2$ in the laboratory test as an initial assessment of $Hg^{II}$ quantitative sorption capacity in order to rule out testing candidate materials that would not be worth pursuing in more expensive and time consuming field deployments (for example, work with PFSA-M was discontinued, lines 126-129). Additional testing with a range of representative $Hg^{II}$ compounds should be performed for any promising new sorptive surface before it can be deployed in the field with confidence, as stated on Lines 305 and 306. Field tests demonstrated that these materials did not collect ambient $Hg^{II}$ quantitatively and thus, testing with additional laboratory salts was not necessary. Referee #1 requested a discussion of how well $HgBr_2$ may represent atmospheric $Hg^{II}$ compounds so the following was included in the Introduction:

**Line 32, page 1:** "A complete understanding of Hg behavior in the atmosphere is necessary to describe the fate of anthropogenic Hg pollution, assess health risks to humans and wildlife, and evaluate the effectiveness of the Minamata Convention. The mechanisms that govern the oxidation and reduction of Hg in the atmosphere are not well understood (Shah et al., 2021), and model results are uncertain, because they do not consider other forms of $Hg^{II}$ present (Gustin et al., 2023). For example, Shah et al. (2021) assumed all $Hg^{II}$ compounds devolatilized from aerosols are $HgCl_2$. This is an assumption that has not been validated.

Currently, Br- and Cl- radicals are considered to participate in elemental Hg ($Hg^0$) oxidation. This is based on both theoretical work (Holms et al., 2010; Horowitz et al., 2017; Song et al., 2024) and experimental observations of atmospheric Hg depletion events in the Arctic (Steffen et al., 2008), as well as observations of $Hg^{II}$ formation at the marine boundary layer (Laurier et al., 2003). The identity of $Hg^{II}_{(g)}$ compounds in the atmosphere is currently unknown, but mass spectrometry (MS) methods capable of observing atmospheric $Hg^{II}_{(g)}$ speciation are in development (cf., Jones et al., 2016; Khalizov et al., 2020; Mao and Khalizov, 2021). These methods have been developed using $HgBr_2$ and $HgCl_2$ as model atmospheric $Hg^{II}_{(g)}$ compounds, given the role of halogen radicals in atmospheric $Hg^0$ oxidation. However, due to differences in $Hg^{II}_{(g)}$ behavior, the use of a broad range of representative compounds is desirable in both MS method development and validation of preconcentration surfaces."

**The text was additionally corrected throughout for grammar and clarity, where appropriate.**

[revised manuscript text omitted]